# Snow stratigraphy observations from Operation IceBridge surveys in Alaska using S/C band airborne ultra-wideband FMCW radar

Jilu Li[1], Fernando Rodriguez-Morales[1], Xavier Fettweis[2], Oluwanisola Ibikunle[1], Carl Leuschen[1], John Paden[1], Daniel Gomez-Garcia[1], Emily Arnold[1]

[1]The Center for Remote Sensing and Integrated Systems (CReSIS), University of Kansas, Lawrence, KS 66045, USA
[2]Department of Geography, University of Liege, Liege, Belgium

*Correspondence to*: Jilu Li (jiluli@ku.edu.com)

**Abstract.** During the concluding phase of the NASA Operation IceBridge (OIB), we successfully completed two airborne measurement campaigns (in 2018 and 2021, respectively) using a compact S/C band radar installed on a Single Otter aircraft and collected data over Alaskan mountains, ice fields, and glaciers. This paper reports seasonal snow depths derived from radar data. We found large variations in seasonal radar-inferred depths with multi-modal distributions assuming a constant relative permittivity for snow equal to 1.89. About 34% of the snow depths observed in 2018 were between 3.2 m and 4.2 m, close to 30% of the snow depths observed in 2021 were between 2.5 m and 3.5 m. We observed snow strata in ice facies, wet-snow/percolation facies, and dry snow facies from radar data and identified the transition areas from wet-snow facies to ice facies for multiple glaciers based on the snow strata and radar backscattering characteristics. Our analysis focuses on the measured strata of multiple years at the caldera of Mount Wrangell to estimate the local snow accumulation rate. We developed a method for using our radar readings of multi-year strata to constrain the uncertain parameters of interpretation models with the assumption that most of the snow layers detected by the radar at the caldera are annual accumulation layers. At a 2004 ice core and 2005 temperature sensor tower site, the locally estimated average snow accumulation rate is ~2.89 m w. e. a$^{-1}$ between the years 2003 and 2021. Our estimate of the snow accumulation rate between 2005 and 2006 is 2.82 m w. e. a$^{-1}$, which matches closely to the 2.75 m w. e. a$^{-1}$ inferred from independent ground-truth measurements made the same year. The snow accumulation rate between the years 2003 and 2021 also showed a linear increasing trend of 0.011 m w. e. a$^{-2}$. This trend is corroborated by comparisons with the surface mass balance (SMB) derived for the same period from the regional atmospheric climate model MAR (Modèle Atmosphérique Régional). According to MAR data, which show an increase of 0.86°C in this area for the period of 2003-2021, the linear upward trend is associated with the increase of snowfall and rainfall events, which may be attributed to elevated global temperatures. The findings of this study confirmed the viability of our methodology, as well as its underlying assumptions and interpretation models.

## 1 Introduction

Glaciers outside Greenland and Antarctica play an important role in the Earth's climate system and respond rapidly to changes in climate which impacts regional hydrology and the local economy. According to a recent report [WCRP Global Sea Level

Budget Group, 2018], these glaciers are the second largest contributor to sea-level rise, after ocean thermal expansion, contributing 21 percent of the global mean sea-level rise during the period between 1993 and 2018. Another study claims that global glaciers are increasingly losing ice mass since the twenty first century and contributed 6 to 19 percent of the observed acceleration of sea-level rise during 2000-2019; the mass loss of Alaska glaciers was the biggest contributor and accounted for 25 percent of the global glacier mass loss compared to the second largest contributor, glaciers of the Greenland periphery, with 13 percent [Hugonnet et al., 2021]. The volume loss of the glaciers in Alaska is ~10 percent of the estimated total mean annual freshwater discharged into the Gulf of Alaska (Neal et al., 2010 and Hill et al., 2015). The glacier discharges affect stream flow and stream temperature that are critical to the spawning and incubation of Pacific Salmon in the Copper River region of the Gulf of Alaska, which is home to important fisheries [Shanley and Albert, 2014].

Snow accumulation on glaciers are key components to understand and model the process of glacier mass loss. Existing spaceborne remote sensing techniques are routinely used to map snow cover extent. However, these observations offer limited capabilities for deriving snow depth and snow water equivalent (SWE). For active microwave sensors, only wet snow can be recognized reliably. However, high-water content severely reduces the signal penetration depth into the snow. Passive microwave sensors can map dry snow, but their spatial resolution is coarse and only ~1 m snow depth can be mapped [Dietz et al., 2012]. For ground-based measurements, snow depth and accumulation are usually estimated using *in-situ* probe and /or snow pit measurements [Benson, 1968; Kanamori et al., 2005; Stuefer et al., 2020], automatic records from snow pillow, temperature sensors and weather stations [Beaumont, 1965; Kanamori et al., 2008], and measurements from ground-penetrating radar (GPR) [McGrath et al., 2015]. Ground-based methods provide detailed measurements about snow properties including temperature, snow grain shape and size, hardness, density, and layer information. The major drawback of ground-based methods is that they are either sparse point observations or only provide very limited continuous spatial coverage. Airborne remote sensing with GPR and FMCW (Frequency-Modulated Continuous Wave) radar has been demonstrated to be a cost-effective method to provide measurements with fine spatial resolutions and comprehensive regional coverage [McGrath et al., 2015; Yan et al., 2017]. Ground-based measurements are used to validate both airborne and satellite observations and data products, and airborne data can also be used to validate satellite observations and data products [Lindsay, et al., 2015; Ramage, et al., 2017; Largeron, et al., 2020; Jeoung et al., 2022].

Direct snow depth and layer measurements at a glacier-scale are rare in Alaska because of their difficult accessibility. Arcone [2002] analyzed data collected in the early summer of 1994 by a helicopter-borne 135 MHz short-pulse radar over the Bagley Ice Field and provided estimates of snow depths and refractive indices based on diffraction and reflection characteristics of snow layers within temperate firn. In the spring of 2013, Gusmeroli et al. [2014] and McGrath et al. [2015] measured snow accumulation of several glaciers around the Gulf of Alaska using 500 MHz ground- and helicopter-based ground-penetrating radar instruments. Complemented by ground truth observations, they showed highly variable SWE over short spatial scales. In the late spring of 2018, Li et al. [2019] collected snow data over Alaskan glaciers, ice fields and mountain caps using a

compact ultra-wideband FMCW radar installed on a Single Otter aircraft. The radar operated at a center frequency of 5 GHz with a 6-GHz bandwidth. They observed seasonal snow depth around the areas of Logan Glacier, Walsh Glacier and upper Hubbard Glacier, as well as deep multiyear snow stratigraphy of the snow caps of Wrangell and Bona mountains (see Fig. 1).

    In the spring of 2021, we had a follow-up airborne campaign in Alaska at the closing of the NASA OIB and observed snow
stratigraphic layers across a broader region than the 2018 campaign [Li et al., 2019]. The objective of this paper is to report our new snow depth observations, derive preliminary SWE values and snow accumulation with the combined 2018 and 2021 airborne radar datasets, and compare the results with previous observations and studies. The paper is organized as follows: Section 1 is the introduction and provides background information; Section 2 describes the data collection and processing activities; Section 3 presents radar observational results, analysis methods and discussions; and Section 4 summarizes the
significant findings and draws conclusions.

## 2 Data collection and processing

    We conducted the 2021 OIB Alaska campaign between May 2, 2021, and May 13, 2021. During this period, we collected ~2 TB of snow radar data over 8 days covering 5315 linear km with an along-track resolution of ~1.3 m. The campaign base (Ultima Thule Lodge), the aircraft platform (Single Otter), antenna installation, and on-board LiDAR and radar are largely the
80 same as in the 2018 campaign [Li et al., 2019]. Table 1 lists the key system parameters for the CReSIS's compact FMCW snow radar system for this field campaign. The details of the on-board LiDAR from the University of Alaska, Fairbanks can be found in [Johnson et al., 2013]. The snow radar's transmit antenna was installed in a protective dielectric radome under the nose of the aircraft, and its receive antenna and the LiDAR were installed in a circular port located in the aft area of the aircraft. In order to be adaptive to the large variations in the altitude above the ground level (AGL) during the flight caused by the
85 complex mountain topography, we operated our snow radar with a 4-GHz bandwidth between 2 GHz and 6 GHz instead of 2-8 GHz as done in 2018 and kept the same chirp length and sampling frequency (see Table 1). This restricts the de-ramped received signals to the first Nyquist zone (<62.5MHz), thereby setting the maximum survey altitude to ~586 m (~1923 ft.) AGL. The bandwidth reduction results in a commensurate degradation in vertical resolution in free space from 2.5 cm to 3.75 cm, but this did not affect the signal-to-noise ratio and snow penetration in a significant manner. Rodríguez-Morales et al.
[2021a] and Li et al. [2019] give the details about the compact snow radar development, key system parameters and the general instrument configuration used onboard the Single Otter aircraft. Additionally, more recent changes and improvements made to the system are documented in Rodriguez-Morales et al. [2021b].

    The two survey regions in Alaska, designated as A and B respectively, are shown in Fig. 1(a) on the hillshade map using the
95 geographic coordinate system NAD83. A is a 4500 km$^2$ area that was only surveyed in 2018. The primary region, B, is an 83,200 km$^2$ area that was surveyed in both 2018 and 2021. The location of Ultima Thule Lodge is indicated by the red star. The flight paths for areas A and B in both years are shown in Fig. 1(b) and (c), respectively. The campaign's flight lines for

2018 are colored green and red, while those for 2021 are colored black and blue. Many of the regions of B that were surveyed in both campaigns overlap. The two-letter annotations, which use the first two letters in their names, identify the locations of the glaciers and mountains discussed in this text. The spatial sampling for a few glaciers (Nabesna Glacier (NA), for example), the Bagley Ice Field (BA) and the snow cap of Mount Wrangell (WR) is denser in 2021 as compared to the flight lines in 2018. This was achieved by using zigzag and gridded flight lines. The new areas surveyed in 2021 include Yahtse Glacier (YA) and Malaspina Glacier (MA) on the coast of Alaska Gulf; Columbus Glacier (CO) and Seward Glacier (SE) on the east of Bagley Ice Field; and Kaskawulsh Glacier (KA) in Canada's Kluane National Park and Reserve.

After the campaign, we first compared the elevation measurements over flat and smooth surfaces with the simultaneous laser measurements and calibrated the radar system delay ($0.064\ \mu s$ and $0.039\ \mu s$ in 2018 and 2021, respectively). We processed the radar data with differential GPS and INS information to improve the geolocation accuracy. Figure 2 shows the data processing flowchart with eight main steps:

1) The GPS and radar data were synchronized using the UTC time stamp stored in the raw radar data files. The accurate longitudes, latitudes, and elevations of the radar phase center along the flight path were computed with the position information of the radar and GPS antenna and the information of aircraft attitudes provided by the onboard IMU (Inertial Measurement Unit) system. Each trace of the raw radar data was tagged with the longitude and latitude of the radar antenna's phase center as its geolocation, and the elevation of the antenna's phase center was used as the zero reference for the two-way travel time (TWTT) from the aircraft to the surface.

2) The coherent noises were automatically tracked by finding the near-DC component in slow-time and were removed by subtraction. Coherent noise was caused by the feedthrough signal due to antenna coupling and undesired spurious signals generated from active microwave components within the radar system. These undesired signal components would reduce the signal-to-noise ratio (SNR), interfere with surface tracking and deconvolution if were not removed.

3) A fast-time FFT (Fast Fourier Transform) was applied trace by trace with a Hanning window to reduce range sidelobes. This step, analogous to pulse compression, obtained the target response as a function of range.

4) A deconvolution filter was applied after the fast-time FFT to further reduce sidelobes and the range resolution degradation due to any other system artifacts, such as small signal reflections between radar hardware components, filters' nonlinear group delays, the digital chirp's amplitude variations and frequency nonlinearity. Minimizing range sidelobe level is important because range sidelobes from strong interfaces could be misinterpreted as snow layers or mask weak reflections from real interfaces. The implemented deconvolution filter was an inverse filter of the radar system impulse response which was derived using specular returns from an electrically smooth surface such as the calm-water surface of a lake.

5) The coherent integration step was performed by stacking data traces together with the averages. This process was an unfocused SAR (Synthetic Aperture Radar) processing to improve the SNR. It included both hardware and software stacking. The hardware stacking was implemented within the radar's digital system and reduced the volume size of

the recorded data. The software stacking was carried out after the deconvolution in data processing. The incoherent integration was carried out after the coherent software stacking by taking the average of the squared data of several traces. Incoherent integration reduced the signal fading effects and the data size of the final radar echogram. The number of traces in the coherent hardware integration was 8 and 16 in 2018 and 2021, respectively. The number of traces in the coherent software and incoherent integrations was 2 and 5 respectively in both 2018 and 2021. The PRF (Pulse Repetition Frequency) was 4000 Hz and 6250 Hz in 2018 and 2021, respectively. The combined coherent and incoherent integrations determined the spatial sampling frequency along the flight path and the along-track resolution depended on the aircraft velocity and the effective PRF which is 50 Hz and 39.0625 Hz in 2018 and 2021, respectively. At the typical velocity of 50 m/s during the surveys, the along-track resolution was 1m and 1.28 m in 2018 and 2021, respectively.

6) The surface was automatically tracked at this step using a threshold method. The automatic tracking usually picked the surface consistently except at the locations where the Nyquist zone changed, or the surface elevation changed very rapidly between narrow valleys. In the latter case the backscattering from both sides appeared in the leading edge of the surface and affected the threshold tracker. At these locations we corrected the surface tracking semiautomatically in our picker using manual control points.

7) The data was elevation compensated with accurately tracked surface to remove large aircraft elevation changes for effective data truncation, display radar echograms and post radar images. The two mostly used compensation options in our processing routine were WGS- 84 elevation compensation and depth elevation compensation. The radar echogram or image was showing the real surface topography in WGS-84 datum after the WGS-84 elevation compensation. The surface was flattened after the depth elevation compensation to better display the depth between snow layers. The depth elevation compensation was implemented by using a low pass filter to get a smoothed version of the tracked surface in radar echograms, the smoothed surface was then used as the zero-depth reference and the radar echograms were normalized to this reference. The high-frequency texture of the surface was therefore kept after the surface flattening.

8) The final processed radar data and images were generated according to selected elevation compensation method.

The same processing steps and parameters were used in processing the 2018 and 2021 datasets except the above-mentioned different bandwidth, hardware stacking and PRF settings. More discussions about the general data processing procedures for the snow radar can be found in [Panzer et al. 2013; Yan et al., 2017].

Unlike the campaigns in Antarctica and Greenland, where open water leads were occasionally available as specular targets for deriving the radar's system impulse response and then using these data for deconvolution, during the two Alaska campaigns, we used data collected over the water surface of lakes by the coast for deconvolution. Supplementary section S1 presents the radar's system impulse response derived using the reflections from the surface of Malaspina Lake during the 2021 campaign,

and the sample radar echograms and A-scopes in S1 show the range sidelobe reduction obtained by means of our deconvolution algorithm.

## 3 Result analysis and discussions

We observed snow layers of seasonal accumulation and multi-year accumulation over a range of surface elevations from 1007 m to 4621 m above sea level. These observations were from ablation areas at lower elevations all the way up to mountain
summits at high elevations. In this section, we first present the overall seasonal snow observations, then focus on the analysis of snow accumulations at the caldera of Mount Wrangell, and lastly discuss the observations along the transition from accumulation to ablation along several glaciers.

### 3.1 Observations of seasonal snow

The red and blue flight lines in Fig. 1(b) and (c) show the locations where we picked the seasonal snow layer in both 2018 and
2021, respectively. This layer may be the earlier old ice in ablation areas or the first distinct layer in accumulation areas. The first distinct layer in accumulation areas may have ambiguity to be the previous summer layer when snow layers exist within the annual layer for deep snow cover. Figure 3(a) presents a radar echogram for a 10-km segment along the main trunk of the east Bagley Ice Field. The red line in the map of Fig. 3(b) shows the geolocation where we retrieved the radar data on May 2, 2021. The glacier surface profile is flattened in Fig. 3(c) to better show the snow depth, which is around 3 m. The surface
elevation of this segment is between 1326 m and 1423 m in the ablation area (see Table 4 and Fig. S5-5). Figure 3(d) and (e) give the distributions of tracked seasonal snow depth for both 2018 and 2021, respectively. Both years have multi-modal peaks largely ranging between 1-6 m. For the 2018 data, the mean values of the three distributions are around 1.2 m, 3.7m and 5.5 m. For the 2021 data, the mean values of the two distributions are around 1.1 m and 3 m. The third distribution in 2018 were mainly from thick seasonal snow along Logan Glacier and the upper Hubbard Glacier where we did not fly over these locations
in 2021 (see Fig. 1(c)). About 34% of the depths observed in 2018 were between 3.2 m and 4.2 m, close to 30% of the depths observed in 2021 were between 2.5 m and 3.5 m. Given the low number of occurrences, we truncated both distributions for depths beyond 8 m. It is noted that there are few locations at high elevations where the seasonal snow depth could be greater than 15 m. For the snow depth calculations, we used a value of 1.89 for the real part of the relative permittivity. This value is from the mean velocity of CMP (common midpoint) and probe measurements at seven glaciers in Alaska, $2.18 \times 10^8$ m/s
[McGrath et al., 2015]. We note that the Bagley Ice Field is a temperate glacier, and previous investigations based on 135-MHz pulsed radar measurements in early summer 1994 determined the relative permittivity from 16.81 to 20.25 for the near-surface of Bagley Ice Field. The values are much higher than 1.89 because the 1994 measurements were taken in the early summer when significant melting and drainage occurred [Arcone, 2002]. There are not many large-scale radar snow measurements over Alaska glaciers, yet such measurements are very important for studies on regional hydrology and mass
balance. The goal here is to present the spatial distributions of the seasonal snow our radar has detected. We keep track of the seasonal snow cover in our datasets to facilitate these studies. However, the focus of this work does not extend to the above-

mentioned hydrology and mass balance studies which necessitate a detailed understanding of the snow density profile and its tempo-spatial fluctuations.

## 3.2 Observations over mountain summits

Snow covers with clear annual layers at high-latitude and high-elevation mountain summit areas contain information about the past climate of the area. Several ice cores were drilled decades ago at the caldera of Mount Wrangell and at the Mount Bona-Churchill saddle to study the local climate history [Benson, 1984; Holdsworth et al., 1992; Goto-Azuma et al., 2003; Fisher et al., 2004; Shiraiwa et al., 2004; Zagorodnov et al., 2005; Yalcin et al., 2006; Urmann, 2009]. Mount Wrangell (62°00'21" N, 144°01'10" W, 4317 m a.s.l.) is a large active shield volcano with an ice-filled caldera extending 4 by 6 km in diameter at its

broad summit. The summit region above 4000 m. a.s.l. is over 3 by 8 km and extends into the dry snow zone. Because of these features, researchers have been drawn to study the glacier-volcano interaction [Benson et al., 1975; Benson et al., 2007; Garry et al., 1989], and ice core and climate records [Benson, 1968; Benson, 1984; Yasunari, et al., 2007; Kanamori et al., 2008; Matoba et al., 2014]. Mount Bona (61°23'08" N, 141°44'55" W, 5040 m a.s.l.) and Mount Churchill (61°25'10" N, 141°42'53" W, 4766 m a.s.l.) are also both ice-covered stratovolcanoes. For the snow cap of Mount Wrangell, Benson [1968] obtained

detailed profiles of the temperature, density, hardness, stratigraphy, snow depth and accumulation using snow pit measurements to 10-m depth taken during the summer of 1961. A more recent study determined the snow accumulation at the summit of Mount Wrangell according to the burial times of temperature sensors during the accumulation period between June 2005 and June 2006 [Kanamori et al., 2008]. Mount Bona is about 3 km to the southwest of Mount Churchill, and the saddle between them covers a 4.2 km by 2.7 km area. The Bona Churchill Ice Core BC1 (460.96 m), drilled to bedrock at the saddle

(61°24' N,141°42' W, 4420 m a.s.l.) in the spring of 2002, is one of the only annually dateable records of extended historical duration to ever be recovered from the northeastern side of the Pacific Basin [Urmann, 2009]. A recent study of stable oxygen isotopes (δ18O) in the ice core revealed a strong connection between isotopes at the BC1 site and western Arctic climate [Porter et al., 2019].

To map the annual snow layers formed in recent years around these two areas, we flew over the Mount Wrangell summit on May 25, 2018, May 3, 2021, and May 9, 2021, respectively; and over the Bona-Churchill saddle along southeast-northwest and southwest-northeast flight lines on May 30, 2018 and May 9, 2021, respectively. Figure 4 shows the data coverage of the above surveys. In this figure, the dots with visible spacing depict the flight lines and the dots without visible spacing mark the locations where subsurface layers were observed; the red and blue dots represent the flight lines of 2018 and 2021, respectively.

As shown by the red dots in Fig. 4(a), we flew only a single path through the caldera center of Mount Wrangell in 2018. The path was from east to west and then repeated from the west to east. In 2021, in addition to repeating the flight path of 2018, we surveyed the whole caldera in grids of 1-km spacing along west-east and north-south flight lines. In Fig. 4(a), the black triangle marks the summit of Mount Wrangell. The red star marks the approximate location of the snow accumulation measurements made in 2005 by using temperature sensors installed on a tower at (61°59'26.88" N, 144°01'32.16" W, 4070.41

m a.s.l.), which is also the 2004 ice core drilling site [Kanamori et al., 2008]. The green star marks the location of the crossover between the 2018 and 2021 flight lines at (61°59'09.24" N, 144°00'24.48" W, 4040.32 m a.s.l.). The two locations marked by the stars correspond to the study sites discussed in this section. As shown in Fig. 4(b), the flight lines cross the Bona-Churchill saddle roughly orthogonally along the southwest-northeast and northwest-southeast directions, respectively. The black and red triangles mark the summit locations of Mount Bona and Mount Churchill, respectively. The red star marks the BC1 ice core

site drilled at the saddle in 2002. The green star annotates the location of the data collection segment used to produce the sample radar echogram given in Fig. 5(d). Figure 5(a) is a radar echogram obtained from data collected by flying from north to south over the summit of the Mount Wrangell and the site of the 2004 ice core and 2005 temperature sensor measurements at the caldera center. Figure 5(b) shows a plot of the flight line (in red) on a map with the ice core location annotated by a blue circle. Figure 5(c) displays the conformable sub-surface strata across the caldera. Figure 5(d) presents a radar echogram

produced from data collected by flying over the Bona-Churchill saddle, showing the dense accumulation layers near the BC1 ice core site. In Fig. 5(c) and (d), the surface profiles are again flattened to display the snow layers; the deepest snow depths observed are ~ 81 m at the 2004 ice core site in the caldera of Mount Wrangell and ~ 50 m near the BC1 ice core site at the Bona-Churchill saddle. At a different location marked by the green circle on the map in Fig. 4(b), the deepest layer observed is ~128 m (see the radar echogram provided in the supplementary section S2). For the depth calculation here, we assume an

effective relative snow permittivity of 2.96 obtained according to the interpretation models at the 2004 ice core site in the caldera of Mount Wrangell as described below.

Because there is no snow pit and ice core data available at the time of the radar measurements, we adopt the following interpretation models [Garry et al., 1989] to estimate the approximate depositional ages of the observed snow layers and the

averaged water equivalent accumulation rate over these depositional ages:

$$\frac{dP}{dz} = \rho g cos\alpha \tag{1}$$

$$\frac{d\rho}{dz} = \begin{cases} m_1\rho^2(\rho_I - \rho/\rho_I) & P \leq P^* \\ m_2\rho^2(\rho_I - \rho/\rho_I) & P > P^* \end{cases} \tag{2}$$

$$\frac{dw}{dz} = -\frac{w}{\rho}\frac{d\rho}{dz} - \Delta \tag{3}$$

$$\Delta(z) = \begin{cases} \Delta_0 & z \leq z_s \\ 0 & z > z_s \end{cases} \tag{4}$$

$$\frac{dt_a}{dz} = \frac{1}{w} \tag{5}$$

$$\frac{dt_z}{dz} = \frac{2\sqrt{\varepsilon}}{c} \tag{6}$$

$$\varepsilon = (1 + 8.5 \times 10^{-4}\rho)^2 \tag{7}$$

We refer to the empirical density-depth profile, the snow density-permittivity profile, and the physical processes and

assumptions underlying the equations as interpretation models. The differential Equations (1)-(3), (5) and (6) describe the

variations of pressure $P$, density $\rho$, downward velocity $w$, depositional age $t_a$, and TWTT from the surface to the depth $t_z$, respectively with depth $z$. In Eq. (1), $g = 9.80 \ m/s^2$ is the gravitational acceleration, and $\alpha$ is surface slope. Equation (2) is a modified version of Benson's model [Benson, 1996] in the form of critical pressure $P^*$ with $m_1 = 16.0 \times 10^{-5} \ m^2/kg$, $m_2 = 4.3 \times 10^{-5} \ m^2/kg$, $P^* = 4.459 \times 10^4 \ Pa$, $\rho(0) = \rho_s = 377.36 \ kg/m^3$, the initial snow density at the surface, and $\rho_I = 917.4 \ kg/m^3$, the ice density; these empirical constants were determined from Greenland measurement but fit well to the Mount Wrangell measurements of firn density [Garry et al., 1989]. In Eq. (3), the initial condition is $w(0) = w_s = \rho_w b_w / \rho_s$, where $w_s$ is the annually averaged volume flux of snow at the surface, $\rho_w = 997 \ kg/m^3$ the water density, and $b_w$ the annual water equivalent accumulation at the surface; the flow divergence $\Delta$ is assumed to be constant as $\Delta_0$ to the stagnation depth $z_s$ and to be zero at greater depths as described by Eq. (4) with $z_s = 150 \ m$. Equation (6) describes the TWTT $t_z$ of the electromagnetic wave through the snowpack, where $c = 2.9979 \times 10^8$ m/s is the velocity of light in free space. Equation (7) describes an empirical law for the effect of firn density on relative permittivity [Robin et al., 1969], where $\rho$ is measured in $kg/m^3$. Equation (7) is similar to the Eq. (1) in [Tiuri et al., 1984], which was verified by laboratory dry snow measurements made at four frequencies at 850 MHz, 1.9GHz, 5.6GHz and 12.6 GHz.

Steady-state conditions are assumed for the coupled equations above. The central region of the summit caldera of Mount Wrangell were thought to be near steady state [Benson and Motyka, 1978] based on repeated surveys showing that the surface elevation remained constant within 1 m from 1965 to 1978 [Bingham, 1967; Motyka, 1983]. By looking at the crossovers of the repeated paths flown in 2018 and 2021, the surface elevation changes are close to zero at elevations ~4100 meters (see supplementary section S3 for details). Therefore, the net surface accumulation is roughly balanced by basal melting and outflow to Long Glacier, and we conclude that the steady-state conditions still hold at the time we took measurements. Based on Fig. S3(c), there is a screw towards more positive differences which implies less snow accumulation in 2021. This is supported by the regional atmospheric climate model MAR (Modèle Atmosphérique Régional) outputs which shows the surface mass balance was 3.1 m w. e. and 2.7 m w. e., respectively in 2018 and 2021.

With given initial conditions, we simultaneously integrate the coupled equations to solve for the depositional ages of the observed snow layers. In the previous study by Garry et al., 1989, $b_w = 1.3 \ m/yr$ and $\Delta_0 = 6.075 \times 10^{-3}/yr$ were used based on surface accumulation and motion measurements made in 1965 [Benson et al. 1975], 1965-1966 [Bingham, 1967], and 1975-1976 [Motyka, 1983]. To consider the surface condition changes since then and the spatial variations, we study the sensitivities of the solved depositional ages to these two parameters and determine their appropriate values using the TWTT of snow layers measured by radar as the constraints (other initial conditions and parameters are the same as used by Garry et al., 1989). This is done by minimizing the following cost function:

$$J = \sqrt{\sum_{i=1}^{N}[(t_{a_{i+1}} - t_{a_i}) - 1]^2} \tag{8}$$

where $N$ is the number of observed layers including the surface with index $i = 1$ and $t_{a_1} = 0$ and here the age $t_a$ is for each of the radar horizons. We thus come up with this cost function by assuming that most observed snow layers are annual accumulation layers and the difference between any two consecutive layers should be close to 1.

We choose a location where the surface slope is zero to illustrate our method. Figure 6(a) and (b) show the 2018 and 2021
radar echograms with the surface tracked by a red line and snow layers tracked by blue lines (the depth axis is plotted with effective relative snow permittivity of 2.89 and 2.96, respectively, estimated from the interpretation models respectively). The snow layers were tracked using semiautomatic methods through the GUI (Graphic User Interface) of our picking tool. Control points were manually placed along each layer and one of the automatic linear interpolation, snake and Viterbi trackers was selected to best track the layer between these control points efficiently. The Viterbi tracker typically tracked the layer of interest
most effectively [Berger et al., 2019]. The red dashed lines in each echogram mark the crossover of the flight lines and the location is ~1.127 km southeast of the 2004 ice core site and the snowfall measurements using temperature sensors [Kanamori et al., 2008]. Figure 6(c) and (d) presents, respectively, the A-scopes at the crossover point and the picked layer images of only 50 traces after the crossover point. The horizontal red lines annotate the TWTT measured by the radar at each picked layer. We performed along-track moving average filtering to display the layers more clearly in Fig. 6(c) and (d) and enumerated the
picked layers using numbers with number 1 representing the surface. These annotation numbers are the layer indices in Eq. (8) and Table 2.

For the given initial conditions and model parameters, we first solve Equations (1)-(7) by integration from the surface to the depth of 100 m to determine the relationship between layer depositional age and TWTT. The blue lines in Fig. 6(e) and (f)
show the model results from the 2018 and 2021data frames, respectively. The depositional ages of the observed snow layers are then determined according to the TWTT from the surface to each layer measured by the radar as shown by the red circles on the top of the blue lines in Fig. 6(e) and (f). The cost function $J$ is computed according to Eq. (8) for a range of $b_w$. For the 2018 data frame, there are 16 layers observed and $b_w$ is increased from 1.3 m/yr to 5.2 m/yr in steps of 0.1 m/yr. For $\Delta_0 = 7.3 \times 10^{-3}/yr$, the variations of the cost function with $b_w$ is shown by the blue line in Fig. 7(a). Because the empirical $\varepsilon = $
$f(\rho)$ law determines the model-based travel velocity of the radar signals in snowpack, and thus the modelled $t_a \sim t_z$ relationship, we also computed the cost function using the following empirical equation [looyenga, 1965]:

$$\varepsilon = [\frac{\rho}{\rho_I}(\varepsilon_I^{1/3} - 1) + 1]^3 \qquad\qquad (9)$$

where $\varepsilon_I = 3.17$ is the relative permittivity of ice. As shown in Fig. 7(b), the relative permittivity difference between the two empirical laws described by Eq. (7) and Eq. (9) is less than 3% from the surface to the depth of 100 m, and the effect of this

difference on the cost function (as shown in Fig. 7(a)) can be ignored. Therefore, all the subsequent analyses here will only consider the results using Eq. (7). According to Fig. 7(a), the best estimate of $b_w$ is ~3.3 $m/yr$. Figure 8 shows the variations of the cost function versus $\Delta_0$ over a range between 0.25 to 2 times of $6.075 \times 10^{-3}/yr$. We can see the values of the cost function do not change much, and the best estimate of $\Delta_0$ is $7.3 \times 10^{-3}$ $/yr$, or 1.2 times $6.075 \times 10^{-3}/yr$. Similarly, we determined the optimal values of $b_w$ and $\Delta_0$ for the 2021 data at the crossover, which turned out to be $3.3\ m/yr$ and $7.6 \times 10^{-3}$ $/yr$; and for the 2004 ice core and 2005 temperature sensor tower site, which turned out to be 3.0 m/yr and $3.9 \times 10^{-3}$ /yr.

Table 2 lists the estimated depositional ages of the 16 tracked layers in the 2018 data frame and 19 tracked layers in the 2021 data frame at the crossover point (with the cost function $J = 1.11\ and\ 1.08$ years respectively), and 21 layers in the 2021 data frame at the 2004 ice core and 2005 temperature sensor tower site (with the cost function $J = 1.33$ years). The closer J is to 1, the more the tracked layers are likely to be annual accumulation layers. J increases when there are intra-annual layers tracked. Because we counted dispositional ages from the surface when the data was collected, there might be a constant offset if the first annual layer was not formed one year ago. However, this offset will not affect the annual accumulation rate estimation. From Table 2, we see that most of the layers at the crossover area are identified as annual layers. The 4$^{th}$, and 11$^{th}$ layers in the 2018 echogram are identified as the accumulation layers between annual layers based on their estimated depositional ages; similarly, the 7$^{th}$ and 14$^{th}$ layers in the 2021 echogram are accumulation layers between annual layers. The repeated radar measurements at the same spot after three years enable us to observe how the snow accumulation layers move downwards. However, there exist some shifts in the estimation of depositional ages of the snow layers between the crossover and the ice core/tower site. The shift increases to ~2 years at the 20$^{th}$ layer. The estimation shifts between different sites are expected, considering the snow accumulation process is very complex and are highly affected by the interplay between complex topography and wind redistribution [Winstral et al., 2002]. The surface of the ice core/tower site has a grade of ~2° while the crossover is at a local valley and the effect of wind redistribution on the snow accumulation is not included in the interpretation models.

Therefore, our purpose in this study is not to estimate the accurate depositional age of each snow layer but rather the average snow accumulation rate over years. The annual accumulation rate $r_a(k)$ is estimated according to

$$r_a(k) = \sum_{z_{k-1}}^{z_k} \frac{\rho(z)}{\rho_w}\ dz \tag{10}$$

where $k$ is the depositional age in integer year, $\rho(z)$ is the model-derived density-depth function and $dz = 0.1\ m$ is the step used in integrating the differential Equations (1)-(3) and (5)-(7). The effects of wind redistribution and other factors resulted in the differences in the TWTT measured by the radar for the same snow layers at different locations and thus the depositional

age estimate. These effects have been partly compensated by the optimal values of $b_w$ and $\Delta_0$, and will be further reduced when we estimate the average accumulation rate over multiple years.

Figure 9(a) presents the annual accumulation rates estimated at the two study sites from the interpretation models with the parameters $b_w$ and $\Delta_0$ constrained by the TWTTs to accumulation layers measured by radar. The blue, green, and red circles in the figure present the annual snow accumulation rate estimates, respectively, at the crossover from the 2018 and 2021 radar frames and at the ice core/tower site from the 2021 data frame. The blue and red solid lines are the linear fitting of the estimates at the crossover and the ice core/tower site, showing annual increases of ~0.022 $m\ w.e.\ a^{-1}$ and ~0.011 $m\ w.e.\ a^{-1}$ at the two sites respectively. The interpretation of the horizontal axis should be noted. For example, the estimate in 2020 implies the annual accumulation between 2020 and 2021. As summarized in Table 3, the depth of the deepest layer $D_{max}$ observed at the crossover in Fig. 6(a) and (b) is 70.78m for the 2018 dataset and 80.78 m for the 2021 dataset, with depositional ages of ~15 and ~18 years, respectively. The deepest layer observed at the 2004 ice core and 2005 temperature sensor tower site is at 78.91 m with the depositional age of ~18.6 years. The effective relative snow permittivity $\varepsilon_{r\_eff}$ in Table 3 is calculated as:

$$\varepsilon_{r\_eff} = \left(\frac{c\ t_{z\_max}}{2D_{max}}\right)^2 \tag{11}$$

where $t_{z\_max}$ is the two-way travel time from the surface to the deepest layer at the depth of $D_{max}$ observed by the radar. At the crossover, the estimated accumulation rate between 2005 and 2006 is 2.97 m w.e.a$^{-1}$; the estimated average accumulation rate for the years between 2003 and 2021 is ~3.10 m w.e.a$^{-1}$. At the ice core/tower site, the estimated accumulation rate between 2005 and 2006 is 2.82 m w.e.a$^{-1}$; the estimated average accumulation rate for the years between 2003 and 2021 is ~2.89 m w.e.a$^{-1}$. We see the estimates of 2.82 m w.e.a$^{-1}$ at the ice core/tower site are very close to the ground-truth value of 2.75 m w.e.a$^{-1}$ (see Table 3) which were estimated from the actual accumulation measurements made between June 3, 2005 to December 8, 2005 with an extrapolation to June 22, 2006 [Kanamori et al., 2008].

In addition to comparing the accumulation rates estimated from our radar data with the limited available ground truth from the temperature sensor measurements, we also compared our results with the surface mass balance (SMB) estimates using the regional atmospheric climate model MAR. The MAR model simulates energy and mass flux between the atmosphere and the snowpack using EAR5 reanalysis outputs as a 6 hourly forcing dataset. As it was run here at high resolution (5 km), it replicated mesoscale meteorological processes more realistically and has been validated with in situ data and remotely sensed data over polar ice sheets such as Greenland Ice Sheets (GrIS). Further details about the model were discussed in [Fettweis, 2007; 2020] and more recently in [Amory et al., 2021]. For our comparison, we used MAR v3.12.1 which provided over 80 climate fields such as density profiles, SMB, etc. at 5km-grid resolution across Alaskan mountains, permanent ice fields, and glaciers. We computed the annual SMB by summing the daily measurements within the same cycle used in estimating the annual

accumulation rates from radar data (May-to-April). The daily SMB was the sum of snowfall and rainfall minus the sublimation, evaporation, and run-off meltwater for each day. Figure 9(b) shows the mean annual SMB over Alaska glaciers between 2016-

2021 using the May-to-April cycle. For comparison, we computed the annual SMB at the crossover and the 2004 ice core/2005 temperature sensor tower sites by synchronizing the radar flight line coordinates and the gridded MAR model output using 2D Delaunay triangulation-based interpolation.

In Fig. 9(a), the blue and red stars present the annual SMB values of MAR results at the crossover and the 2004 ice core/2005

temperature sensor tower sites, respectively. The blue and red dashed lines are the linear fitting of these SMB values at the two sites, showing both annual increases of ~0.013 m w.e.a$^{-1}$. At the ice core/tower site, the MAR SMB between 2005 and 2006 is 2.86 m w.e.a$^{-1}$ compared to the estimated accumulation rate from radar data, which is 2.82 m w.e.a$^{-1}$. Figure 9(c) presents the differences between the annual accumulation rate $r_a$ estimated from radar data and the SMB computed from MAR outputs, in which the black dashed line with stars shows the site-averaged differences. The absolute values of the site-averaged

differences are less than 0.27 m w.e.a$^{-1}$ before 2015 and the maximum site-averaged difference is 0.58 m w.e.a$^{-1}$ in 2016. The linear increasing trend from MAR data was almost the same as what was inferred from radar data between 2003 and 2021, although the MAR results have larger variations from year to year, especially after 2015. This linear increasing trend and apparent larger temporal variability in MAR versus radar-based estimates are linked to the increase of rainfall events as a result of global warming (see the increase of 0.86°C in 19 years in this area over 2003-2021 in Fig. 9(d) according to MAR). This

SMB variability driven by the presence of liquid water into the snowpack is smoothed in the radar retrieved signal due to the snowpack compaction and its ability of fully retaining the liquid water. According to MAR, the recent increase of SMB over 2003-2021 is 88% driven by the increase of snowfall accumulation and 12% by the mass gained by rainfall (that is fully retained by the snowpack). The increase of rainfall exceeded the interannual variability, and thus is more statistically significant while the increase of SMB and snowfall are within the interannual variability (see supplementary section S4 for details).

According to MAR data, the surface density in Mount Wrangell's caldera is 317.50 $kg/m^3$. This figure is 16% less than the value we used in the study, 377.36 $kg/m^3$. The models' and accumulation estimations' sensitivity to the surface density values was therefore further evaluated. The discrepancies in the density-depth profiles for the two surface density values are depicted in Fig. 10(a). As seen in Fig. 10(b), as depth is increased, the projected depositional ages for the tracked layers would get less

due to the lower surface density. As opposed to 18.6 years for 377.36 $kg/m^3$, the age of the deepest monitored layer is 17.10 years for 317.50 $kg/m^3$. The variations between the annual accumulation estimates are compared in Fig. 10(c). Although there are some variations in the annual accumulation rate within a given year, the linear increasing trend is nearly the same ($0.011\ m\ w.e.\ a^{-2}$ for 317.56 $kg/m^3$ against $0.012\ m\ w.e.\ a^{-2}$ for 377.36 $kg/m^3$). This makes sense given that, for a lower snow density, the snow mass likewise decreases as the age difference between two snow layers narrows. As a result, we

deduced that the linear upward trend in the annual accumulation rate seen between 2003 and 2021 is not affected much by the surface density.

Table 3 summarizes the comparisons among the ground truth, radar, and MAR results. This is the first time that airborne radar observations, temperature sensor measurements on the ground and MAR outputs have been compared to validate annual snow accumulation over Alaska glaciers where MAR has been applied for the first time with success. We believe that the significant finding of a linear rising trend in accumulation rate between 2003 and 2021 may aid in more precisely estimating the mass loss of Alaskan glaciers and their impact to sea level rise.

3.3 Observations along glaciers

Distinct zones or glacier facies exist for ice sheets and glaciers. These facies, which relate to snow accumulation and ablation, range in elevation from high to low and include dry snow facies, percolation facies, wet snow facies, and ice facies [Benson, 1996]. For instance, the two research sites in Section 3.2 near Pit 5 in [Benson, 1968] are on the dry-snow line and represent dry snow facies since we did not observe internal layer melt from radar echograms. Large scale monitoring of glacial facies provides useful information for hydrological planning (particularly in areas where glacier-fed melt is a significant contributor to total runoff) and potentially early detection of climate changes. Multi-temporal ERS-1 satellite SAR data of 1992-1993 revealed the dry-snow facies, combined percolation and wet-snow facies, ice facies, transient melt areas and moraine [Partington, 1998]. In Partington's study over the area between the north-east slopes of Mount Wrangell and Nabesna Glacier, the elevation of the snowline was around 2100 m, the dry snowline was at elevations around 3460 m, and the combined percolation and wet snow facies were within elevations between 2100 m and 3460 m. We also flew over this same area during our 2018 and 2021 surveys, and observed the strata in dry snow facies, the combined percolation and wet snow facies, and ice facies. Having presented sample radar echograms for the dry snow facies in section 3.2, here we present a sample radar echogram in Fig. 11(a) for the combined percolation and wet snow facies. Figure 11(b) is a plot of the flight line (in red) on the map to show the geolocations of the data, collected on May 25, 2018. The details of the strata are not very clear in this radar image because it is greatly compressed (over a long distance of ~30 km), resulting in low pixel resolutions. Therefore, we also present an image of higher pixel resolutions in Fig. 11(c) for the portion enclosed by the two vertical blue lines in Fig. 11(a) to enhance the granularity of features in the observed strata. We notice that in both images there are some discontinuous layers between the surface and previous summer layer. These internal reflections are roughly parallel with the surface, and the intensity is higher at lower elevations. The melting and refreezing along with pooling of liquid water at storm layer interfaces, which are occasional in the percolation and snow wet facies, might result in these reflections. The snow depth of the previous summer surface shows a high correlation with the glacier surface elevation which decreases from 2815 m to 1943 m as shown in Fig. 11(d), i.e., the annual snow depth increases with elevation.

The boundaries between different glacier facies can be identified according to the stratigraphic features of subsurface layers and C-band radar backscattering signatures [Partington, 1998; Langley et al., 2008; Ramage et al., 2000]. There are many cases in our airborne radar observations where the snowline defined as the boundary between the ice facies and wet-snow facies can

be clearly identified. Figure 12 presents such an example for Kaskawulsh Glacier where the data was collected on May 10, 2021 over a distance of 15 km along the glacier's central line. The glacier's surface profile in the image is flattened to better show the snow layers in Fig. 12(a), and the WGS84 surface elevations of the glacier are between 1913.43 m and 2362.18 m as shown in Fig. 12(c). The snowline location at 6.973 km is marked by the red vertical line in the radar echogram according to the following features observed: 1) the previous summer surface (PSS) is distinct because of its high coherent reflections at most elevations; 2) multiple snow layers are visible at elevations higher than the elevation of the snowline, and these layers converge towards the snowline; 3) at elevations lower than the elevation of the snowline, the PSS is the only visible layer beneath the surface and the backscattering is lower due to the lack of internal scattering sources. In the zone of ice facies, the PSS is the major source of backscattering, while in the zone of wet-snow facies, the backscatter sources include multi-year accumulation layers and volume scattering. The blue line in Fig. 12(b) gives the column-wise-averaged power in the rectangular box at the bottom of Fig. 12(a), and the two horizontal blue dashed lines in this figure at -1.76 dB and 4.81 dB, present the total averaged backscattering powers of the ice facies and wet-snow facies in the boxed region. The orange line in Fig. 12(b) gives the roll angles to show that the power peak in the ice facies was caused by off-nadir backscattering from the surface when the aircraft rolled about 11.7° to the right. The latitude and longitude of the snowline location are respectively 60.6970° and 139.3633°, and the surface elevation of the snowline is 2105.55 m, as marked by the red circle in Fig. 12(c).

Table 4 summarizes the snowline locations and elevations identified from the 2019 data for Kaskawulsh, Steller, Logan, Nabesna glaciers and the east Bagley Ice Field. The last column of the table lists the CReSIS data frames that show the transition from the wet-snow facies to the ice facies. The supplementary section S5 gives the corresponding echograms.

**4 Summary and conclusions**

The major efforts and contribution from the studies presented in this paper include:

1) Successful collection of snow data using CReSIS S/C band compact radar during two field campaigns in Alaska in 2018 and 2021, respectively; the completion of the data processing, and identification of the seasonal snow accumulation layer. The seasonal snow depths have multi-modal distributions. About 34% of the depths observed in 2018 were between 3.2 m and 4.2 m, close to 30% of the depths observed in 2021 were between 2.5 m and 3.5 m.

2) Observation of snow strata in ice facies, wet-snow/percolation facies and dry snow facies, and identification of the wet-snow to ablation transition areas of several glaciers based on the features of snow strata and radar backscattering characteristics.

3) Development of a method to estimate the average snow accumulation rate at the caldera of Mount Wrangell. This method uses the radar observations of multi-year strata to constrain the uncertain parameters of interpretation models based on the assumption that most of the snow layers at the caldera observed by the radar are annual accumulation layers. The estimated snow accumulation rates are very close to the ground truth obtained at the 2004 ice core and

2005 temperature sensor tower site. The noteworthy discovery of the linear rise trend in accumulation rate between the years 2003 and 2021 was corroborated by comparisons with the SMB derived for the same period from the MAR model and may be attributed to elevated global temperatures. The findings of this investigation confirmed the validity of our technique and the assumptions and interpretation models it was based on. Future research may extend these findings throughout the entire caldera for the geographical pattern of snow accumulation utilizing gridded observations of strata.

4) Release of the S/C band snow data we collected in the two campaigns in Alaska as part of NASA Operation IceBridge Mission. These datasets are valuable for hydrology, glaciology, and radar backscattering studies.

*Data availability*. The radar data products are available at https://data.cresis.ku.edu/data/snow/2018_Alaska_SO/ and https://data.cresis.ku.edu/data/snow/2021_Alaska_SO/; they are also available at NSIDC at https://nsidc.org/data/IRSNO1B/. The traced seasonal snow thickness data is available at https://data.cresis.ku.edu/data/misc/Alaska_seasonal_snow/. The data from MAR simulations performed by XF is available at ftp://ftp.climato.be/fettweis/MARv3.12/Alsaka/ (last access: 21 October 2022).

*Author contributions*. All authors contributed to this work. JL participated in the radar installation and field campaigns, collected, and processed the radar data, performed the analysis, and led the writing of the manuscript. FRM contributed to the radar system design, led its implementation and improvements between field seasons, participated in the field campaigns and collected the radar data. XF performed MAR simulation and assisted in MAR data analysis and interpretation. OI assisted in MAR data analysis. CL led the development of radar's digital back-end and contributed to the radar system design and project management. JP contributed to the processing toolbox for the radar data. DGG led the development of the microwave chirp generator used in the radar and contributed to the radar system design. EA contributed to the design and integration of the antenna setup for the radar system and participated in the 2018 campaign for antenna integration into the Single Otter aircraft.

*Competing interests*. The authors declare that they have no conflict of interest.

*Acknowledgements*. We would like to thank all faculty, staff, and students at CReSIS who contributed to the development and improvements of the compact snow radar and supported the deployment. We would also like to thank the Single Otter pilot, P. Claus, who safely supported our surveys with his decades-long flying experience over Alaska. We gratefully acknowledge the support from the staff at Ultima Thule Lodge and Drs. C. Larsen and J. Holt as well as their teams, who provided help during the installation and de-kit of our instrument before and after the campaigns. We especially thank Dr. C. Larsen for kindly providing us the differential GPS, INS and LiDAR data, and the hillshade map used in this paper. Fieldwork and data processing efforts were supported by NASA, under grant NNX10AT68GT.

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

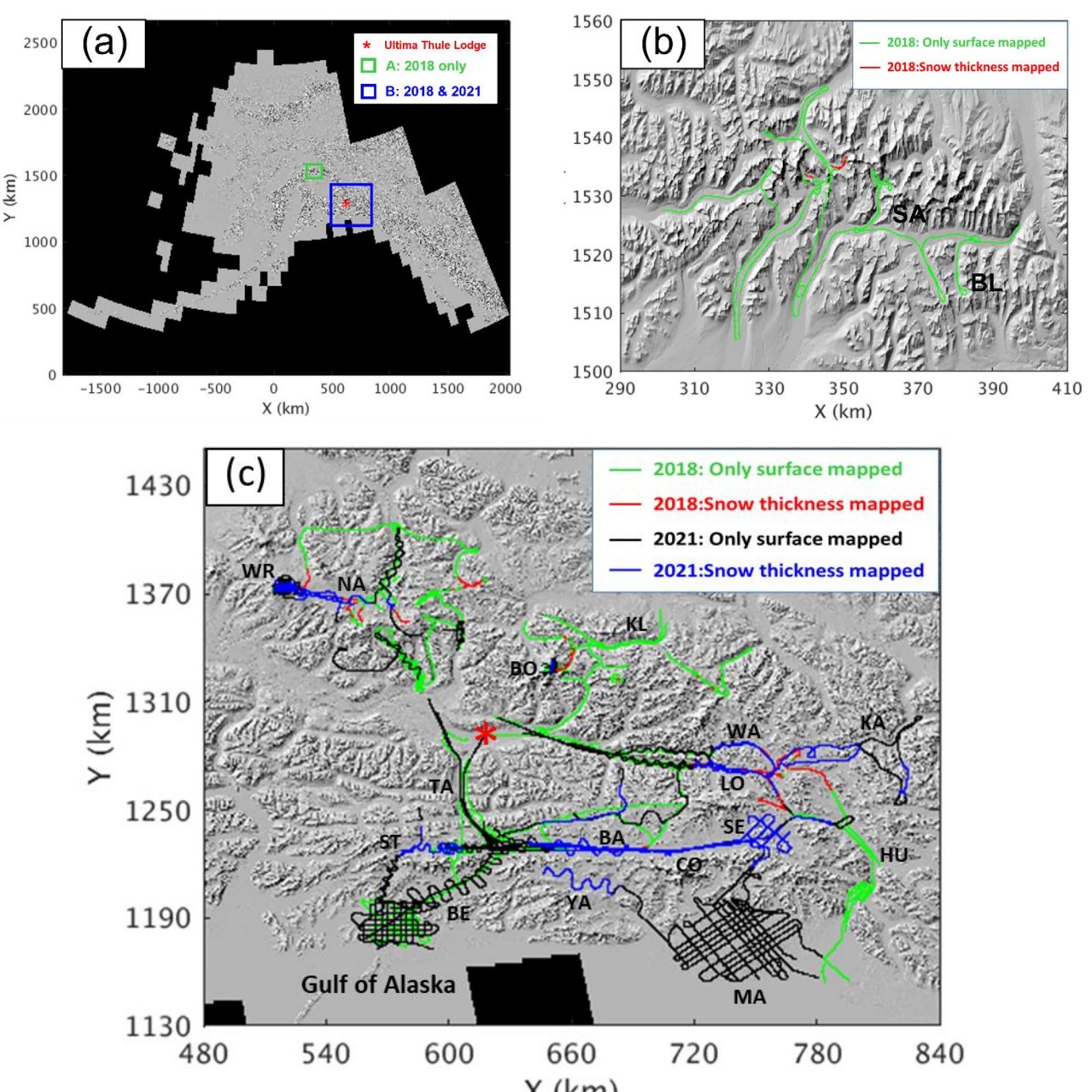


**Figure 1: Coverage maps of Snow Radar data from the OIB surveys in Alaska: (a) locations of survey area A and B; (b) Flight lines over A, surveyed in 2018 only; (c) Flight lines over B, surveyed both in 2018 and 2021. Green and red colors represent the locations where the Snow Radar collected data in 2018; flight lines in black and blue colors represent the locations where the Snow Radar collected data in 2021; specifically, the red and blue lines represent the locations where snow layer or snow-ice interface or snow-**
**rock interface below the surface were observed by the compact Snow Radar. The red star marks the location of Ultima Thule Lodge. The two-letter annotations indicate the locations of some glaciers and mountains using the first two letters in their names. Refer to Fig. 4(a) and (b) for detailed flight lines at Mount Wrangell (WR) and Mount Bona (BO) summits. The hillshade map was provided by Dr. C. Larsen.**

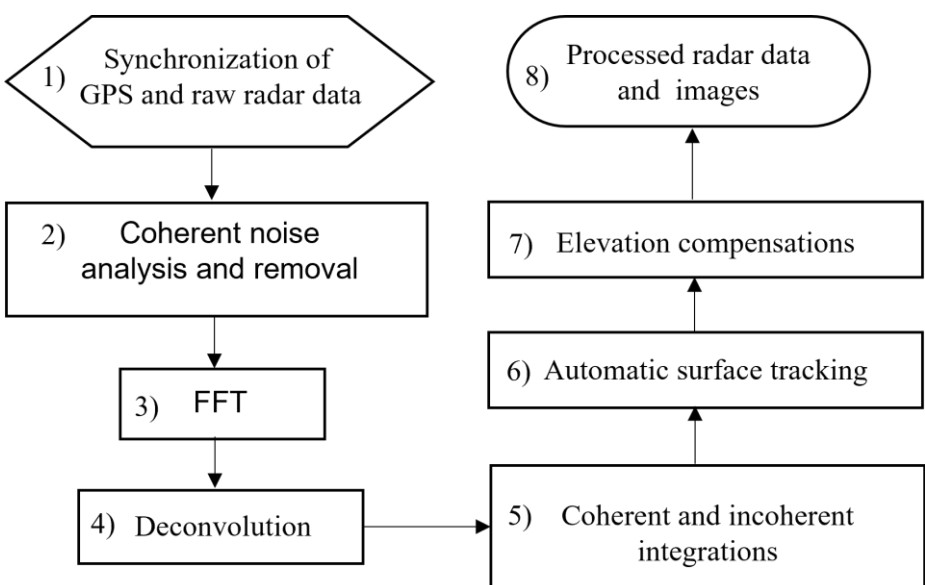

Figure 2: Flowchart of main data processing steps

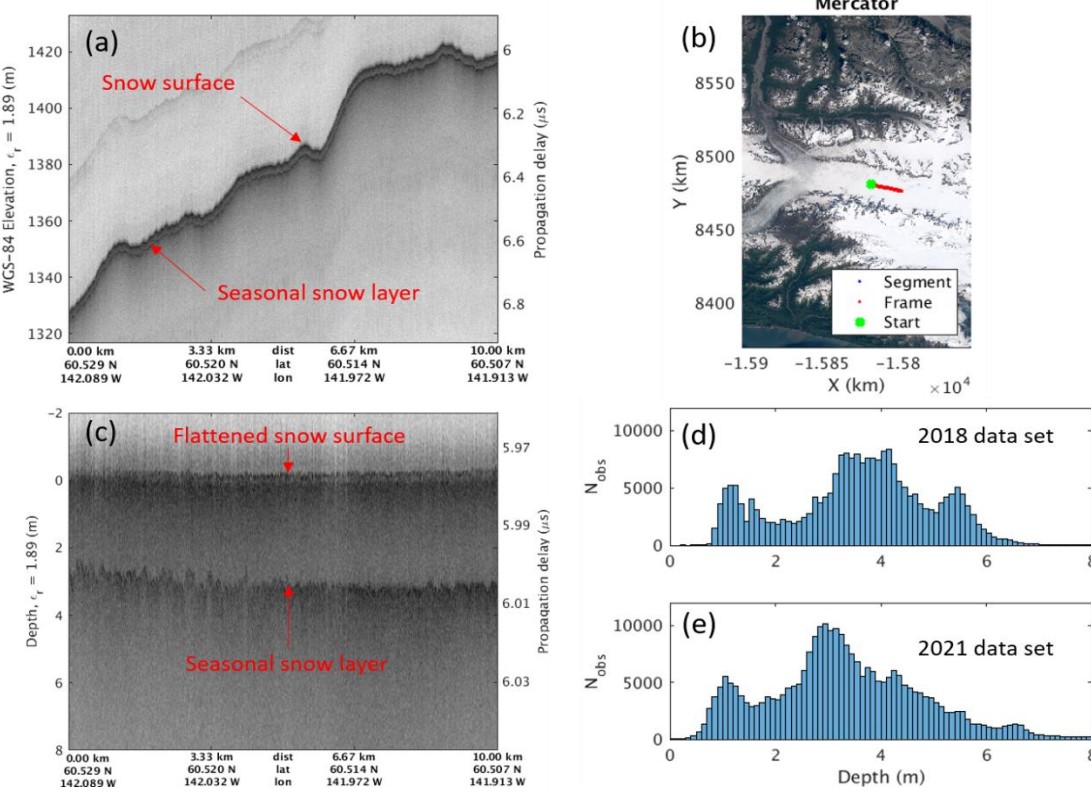

Figure 3: (a) Seasonal snow observations in Bagley Ice Field (BA); (b) Geolocations of the radar echogram indicated by the red line on the Landsat image map; (c) Snow depth around 3 m shown after the glacier surface profile is flattened; (d) and (e) Tracked seasonal snow depth distributions of 2018 and 2021 datasets.

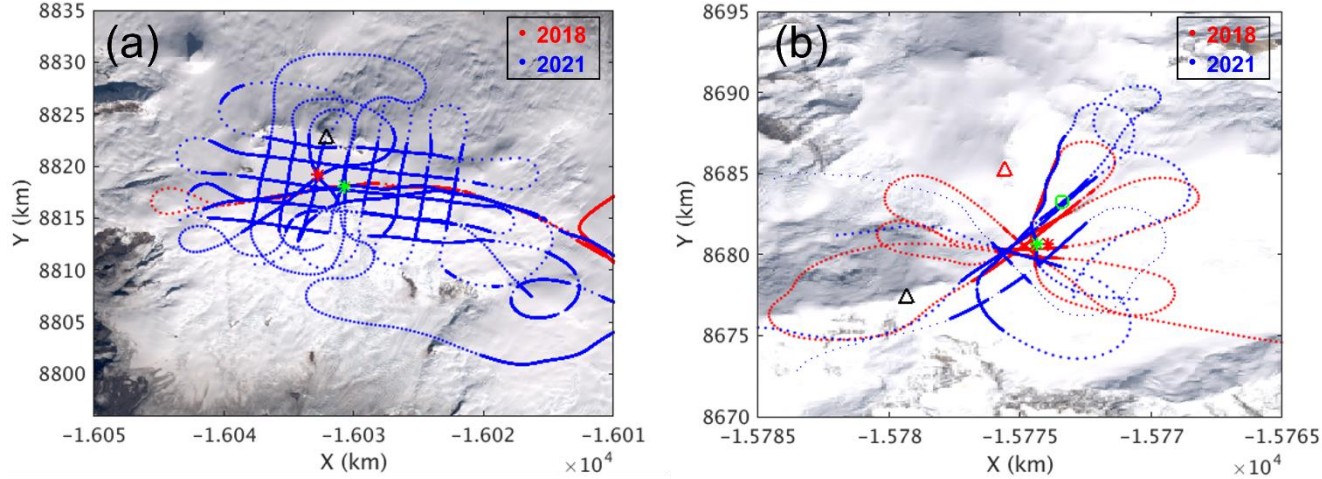

**Figure 4: Data coverage over (a) Mount Wrangell and (b) Mount Bona summit areas plotted on the Landsat image maps. The dots with visible spacing depict flight lines and the dots without visible spacing represent the locations with snow layer/snow-ice interface/snow-rock interface observations. The red and blue dots represent the flight lines of 2018 and 2021 respectively.**


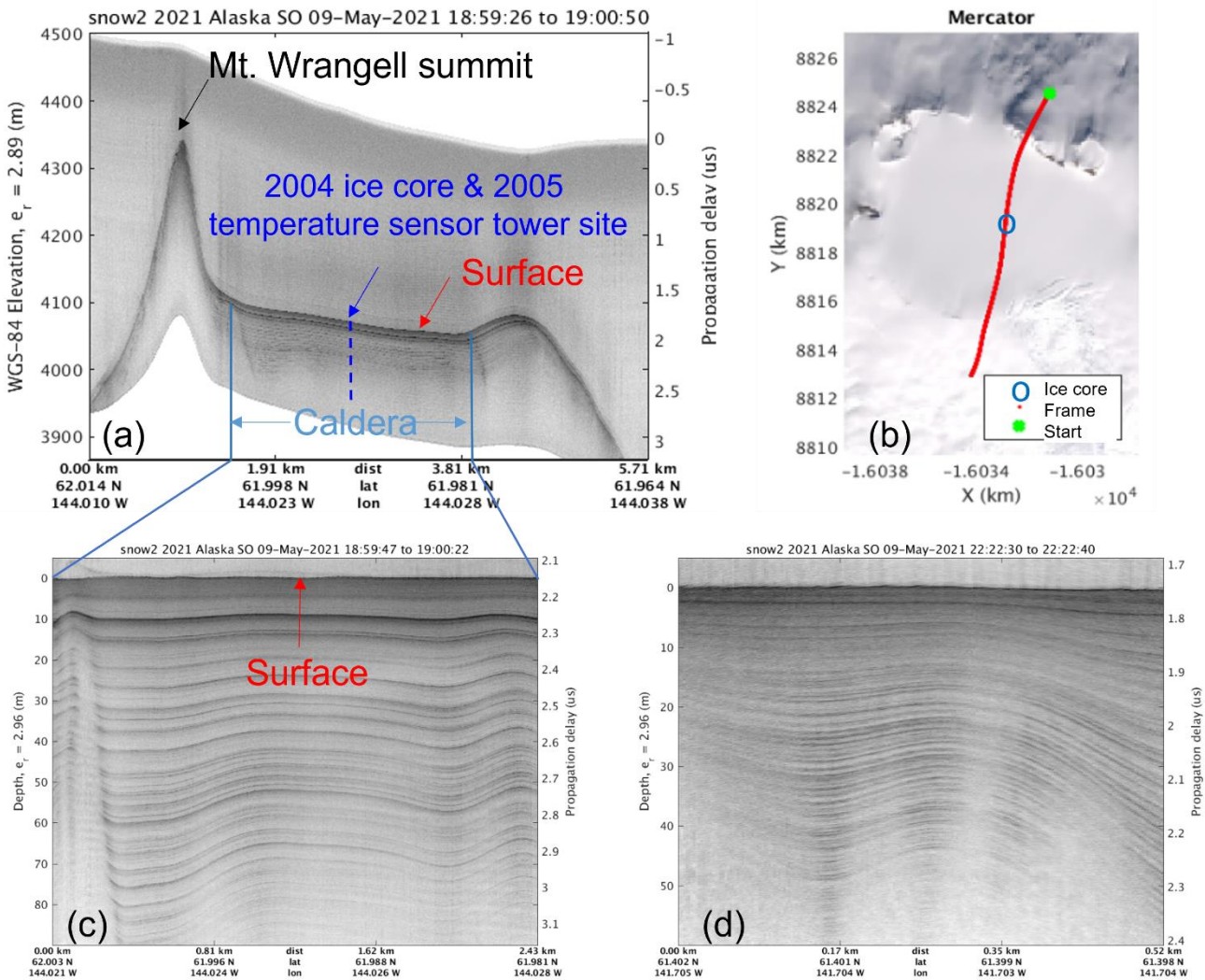


**Figure 5: Conformable snow layers observed at mountain summit areas. (a) Mount Wrangell (WR), data collected on May 9, 2021; (b) the flight line in red annotates the geolocations of the echogram in (a) on the Landsat image map; (c) Accumulation layers in the Caldera; and (d) Bona-Churchill (BO), data collected on May 9, 2021; its location, marked by the green star in Fig. 4b, is close to the 2002 BC1 ice core drilling site. The snow surface profiles in (c) and (d) are both flattened for to better display the layers.**

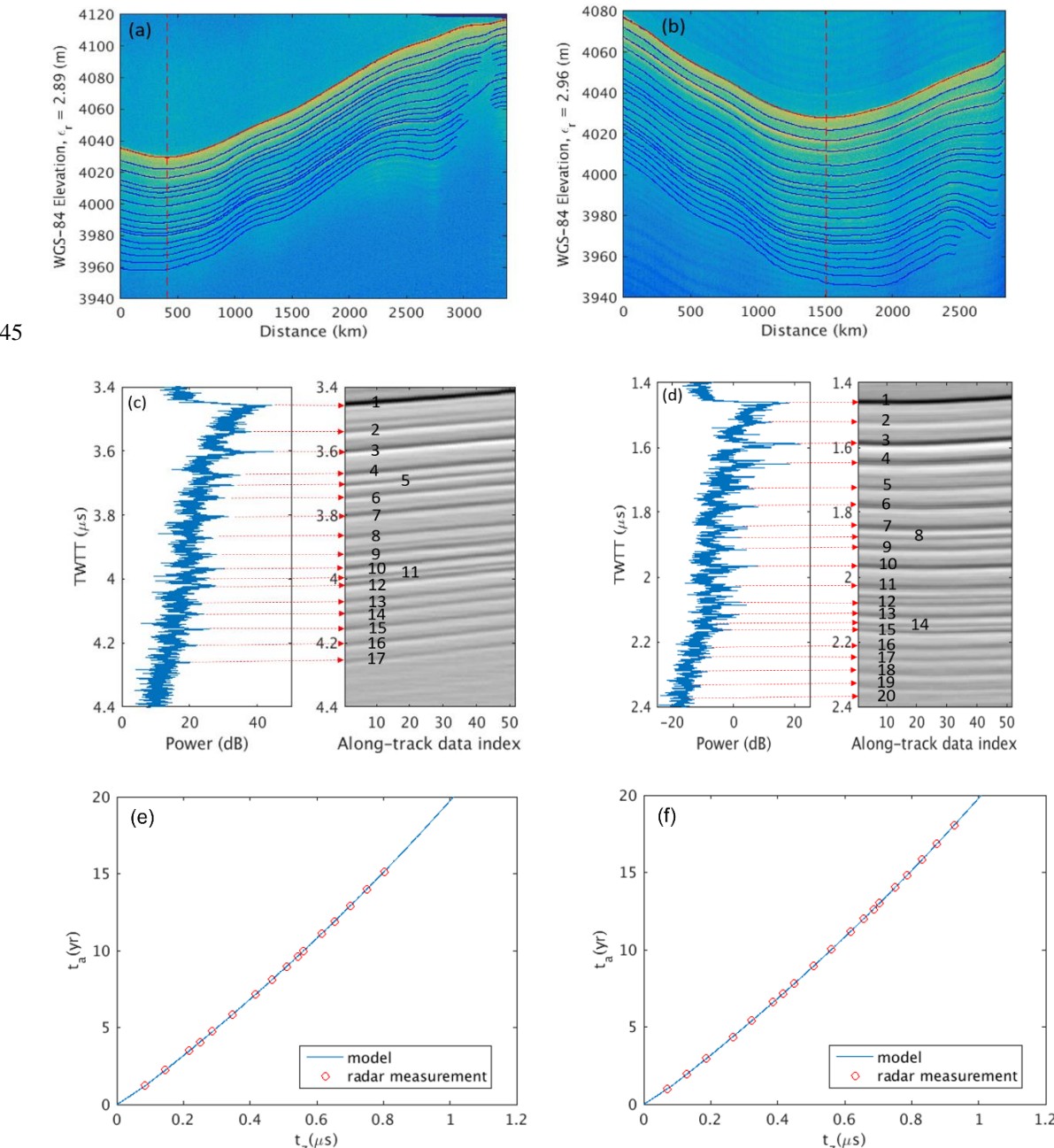

Figure 6: (a) Tracked snow layers using the 2018 dataset (plotted with $\varepsilon_r$ = 2.89 based on the mean velocity between the surface and the depth of 70.80 m at the crossover marked by the vertical dashed red line); (b) Tracked snow layers using the 2021 dataset(plotted with $\varepsilon_r$ = 2.96 based on the mean velocity between the surface and the depth of 80.78 m at the crossover marked by the vertical dashed red line); (c) On the left is the A-scope at the crossover in 2018; On the right are the picked layers marked by sequence numbers after along-track filtering; (d) On the left is the A-scope at the crossover in 2021; On the right are the picked layers marked by sequence numbers after along-track filtering; (e) and (f) Snow layer ages based on model and radar measurements of the TWTTs between the surface and the tracked layers at the location of the vertical red dashed line, from the 2018 and 2021 data frames, respectively.

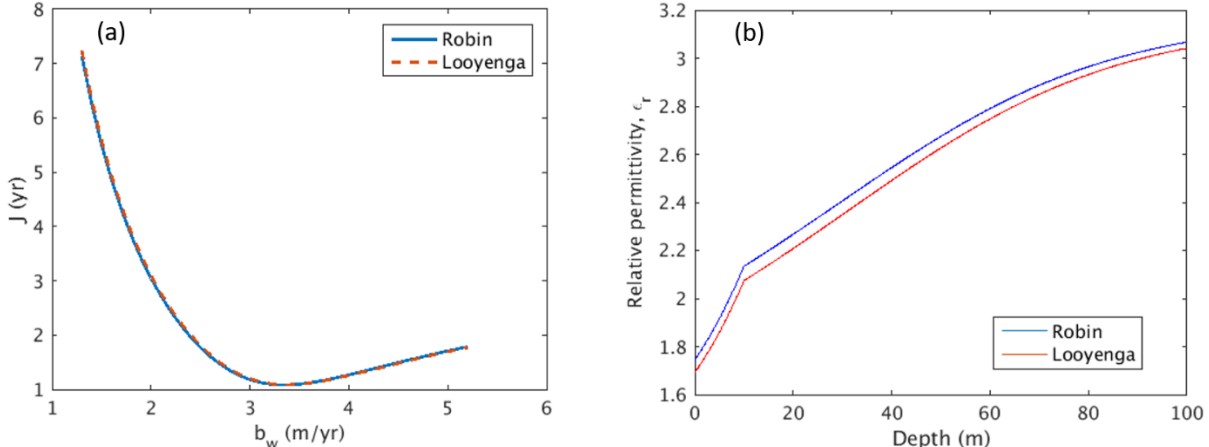

**Figure 7: (a) Cost function of layer ages versus water equivalent surface balance; (b) Empirical density-permittivity models**

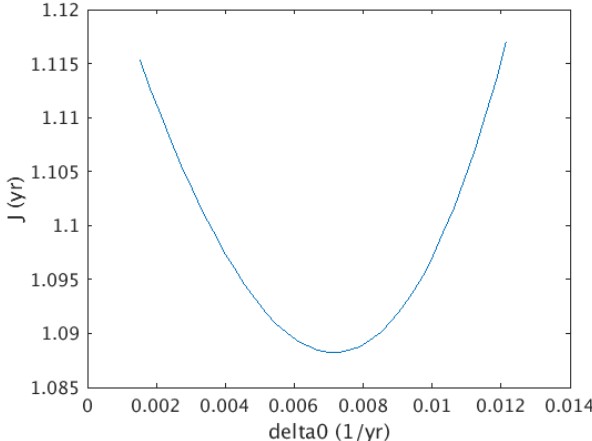

**Figure 8: Cost function of layer ages versus $\Delta_0$**

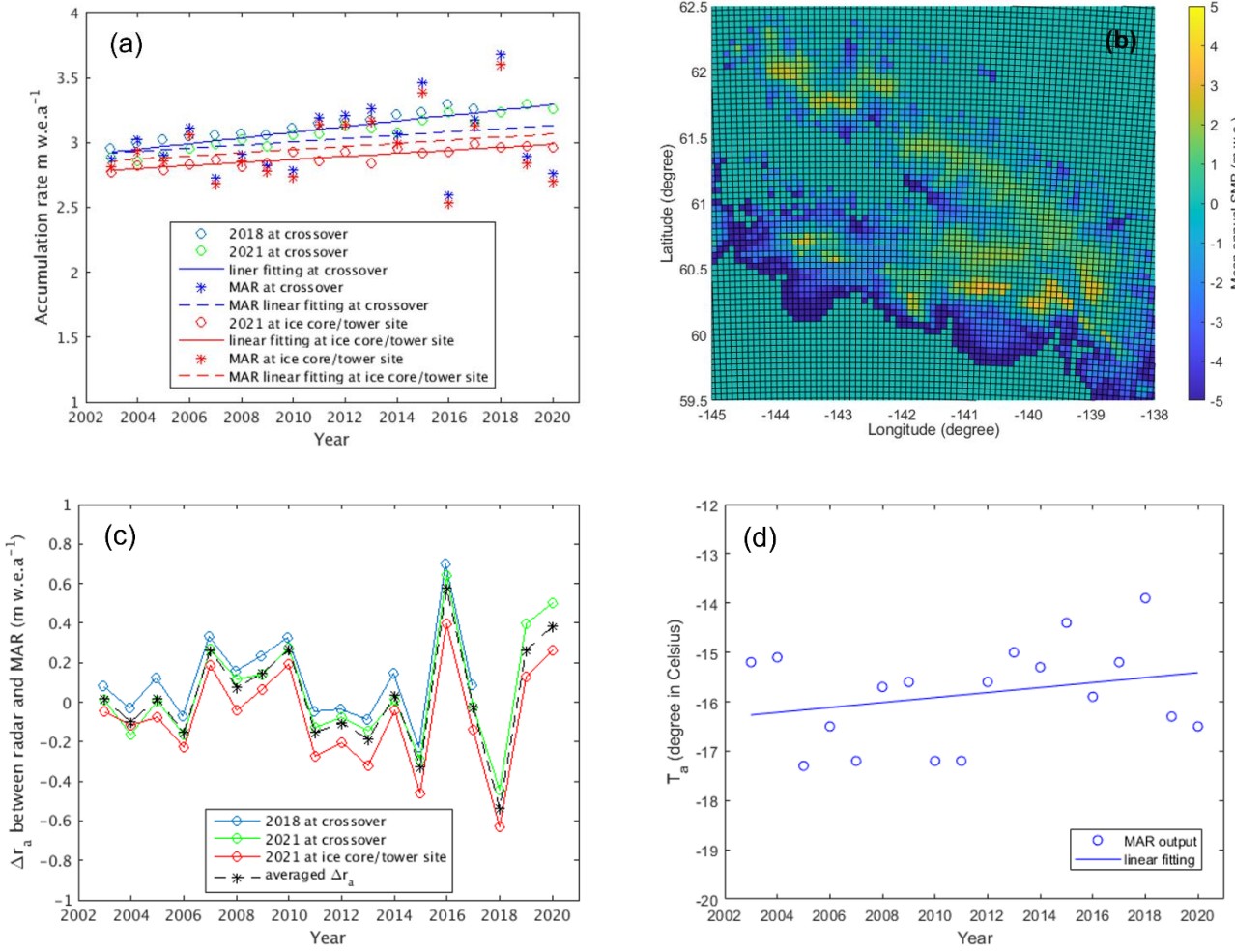

**Figure 9: (a) Estimated annual accumulation rates; (b) MAR map of mean annual SMB over Alaska glaciers between**
**2016-2021; (c) Differences between $r_a$ from radar data and SMB from MAR; (d) Averaged annual temperature from MAR.**

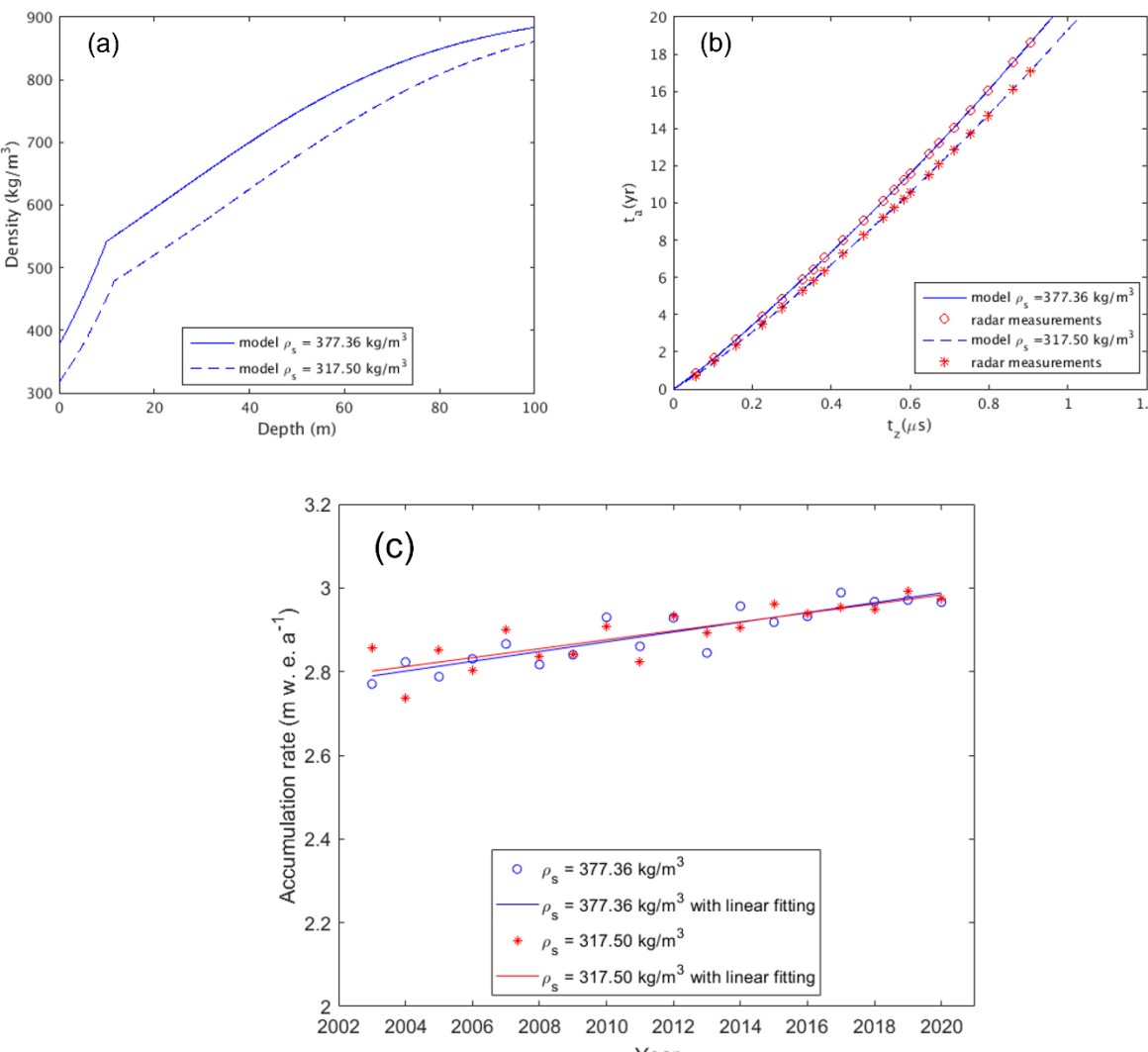

**Figure 10: Depth-density profiles (a), Snow layer depositional ages (b), and estimated annual accumulation rates (c) for two different surface density values.**

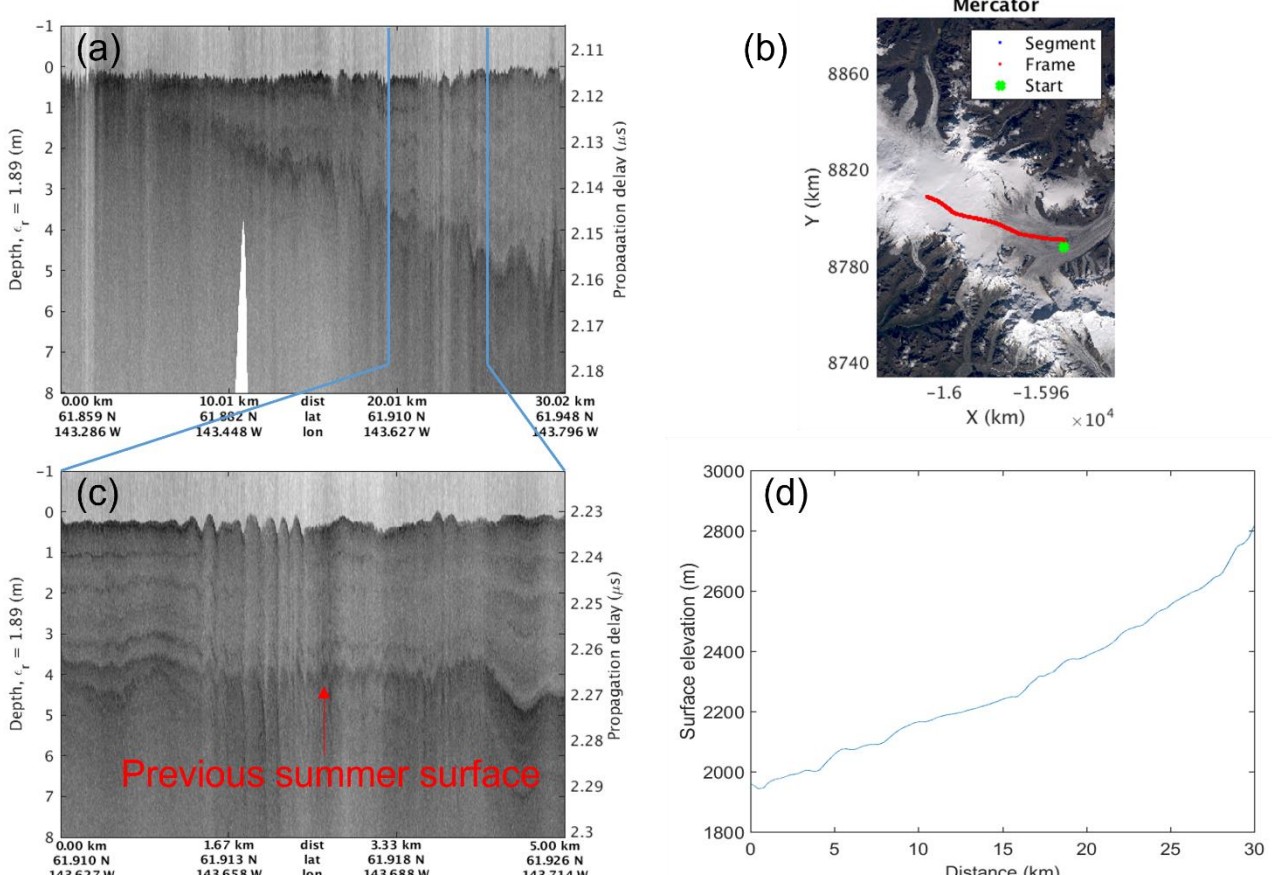

**Figure 11: (a) Observations from the east of Mount Wrangell to Nabesna Glacier; (b) The flight line plotted in red on the Landsat image map; (c) The radar echogram image with clear strata details for the portion between 20 km and 25 km in (a); (d) The surface elevation profile along the flight line.**

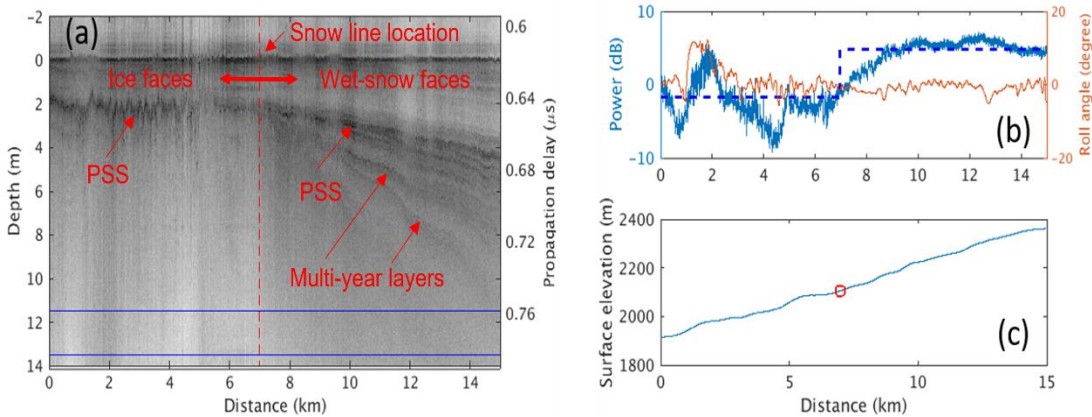

**Figure 12: Snow stratigraphic features during the transition from wet-snow facies to ice facies**


**Table 1: System parameters in 2021**

| Parameter | Value | Units |
| --- | --- | --- |
| Weight (main chassis) | ~35 | lb |
| Dimensions (WHD) | 14.5,9,17 | inch |
| Frequency band | 2-6 | GHz |
| Pulse duration | 250 | μs |
| Pulse repetition frequency | 6.25 | kHz |
| Peak transmit power | ~300 | mW |
| Range resolution | 3.75 (free space) | cm |
| Hardware averages | 16 | |
| Antenna type (TX/RX) | Horn | |
| bandwidth | 2-18 | GHz |
| beamwidth | 86-19/52-9 | degrees |
| gain | 7-13/9-23 | dBi |
| A/D converter | 14 | bit |
| Sampling rate | 125 | MSPS |



**Table 2: Estimated depositional ages of tracked layers**

| | Crossover of 2018 & 2021 datasets $(b_w = 3.3\ m/yr,\ \Delta_0 = 7.3/7.6 \times 10^{-3}\ /yr\ \&\ J = 1.11/1.08\ yr\ for\ 2018/2021)$ | | | | | | | | | | |
|---|---|---|---|---|---|---|---|---|---|---|---|
| 2018 | Layer index $i$ | 2 | 3 | 4 | 5 | 6 | 7 | 8 | 9 | 10 | 11 | 12 |
| | $t_a$ (yr) | 1.19 | 2.20 | 3.43 | 4.00 | 4.67 | 5.77 | 7.06 | 8.04 | 8.88 | 9.52 | 9.87 |
| | Layer index $i$ | 13 | 14 | 15 | 16 | 17 | | | | | | |
| | $t_a$ (yr) | 10.99 | 11.78 | 12.78 | 13.86 | 15.03 | | | | | | |
| 2021 | Layer index $i$ | 2 | 3 | 4 | 5 | 6 | 7 | 8 | 9 | 10 | 11 | 12 |
| | $t_a$ (yr) | 0.98 | 1.92 | 2.9 | 4.29 | 5.32 | 6.51 | 7.09 | 7.74 | 8.83 | 9.90 | 11.09 |
| | Layer index $i$ | 13 | 14 | 15 | 16 | 17 | 18 | 19 | 20 | | | |
| | $t_a$ (yr) | 11.89 | 12.52 | 12.89 | 13.93 | 14.68 | 15.70 | 16.73 | 17.96 | | | |

| | 2004/ 2005 ice core & temperature sensor tower site $(b_w = 3.0\ m/yr,\ \Delta_0 = 3.9 \times 10^{-3}\ /yr\ \&\ J = 1.33\ yr)$ | | | | | | | | | | |
|---|---|---|---|---|---|---|---|---|---|---|---|
| 2021 | Layer index $i$ | 2 | 3 | 4 | 5 | 6 | 7 | 8 | 9 | 10 | 11 | 12 |
| | $t_a$ (yr) | 0.85 | 1.67 | 2.64 | 3.92 | 4.86 | 5.88 | 6.43 | 7.04 | 8.00 | 9.08 | 10.13 |
| | Layer index $i$ | 13 | 14 | 15 | 16 | 17 | 18 | 19 | 20 | 21 | 22 | |
| | $t_a$ (yr) | 10.72 | 11.24 | 11.59 | 12.63 | 13.22 | 14.05 | 14.98 | 16.02 | 17.54 | 18.60 | |

\* Refer to the layers marked by the numbers in Fig. 5 (c) and (d) for the layer index $i$.

**Table 3: Maximum layer depth observed $D_{max}$, effective snow relative permittivity $\varepsilon_{r\_eff}$ and accumulation rates $r_a$ estimated at the two study sites.**

| | 2004/ 2005 ice core & temperature sensor tower site (61.9908°N, 144.0256°W) | | | 2018/2021 crossover (61.9859° N, 144.0068°W) | |
|---|---|---|---|---|---|
| $D_{max}$ (m) | 78.91 | | | 70.78/80.78 | |
| $\varepsilon_{r\_eff}$ | 2.96 | | | 2.89/2.96 | |
| $r_a$ (m w. e. $a^{-1}$) | Radar | MAR | Temperature sensor | Radar | MAR |
| 2005-2006 | 2.82 | 2.86 | 2.75 (ground truth) | 2.97 | 2.90 |
| 2003-2021 (averaged) | 2.89 | 2.96 | NA | 3.10 | 3.03 |
| Linear trend (m w. e. $a^{-2}$) | 0.011 | 0.012 | NA | 0.022 | 0.013 |

**Table 4: Snowline locations of glaciers and ice fields**

| Glacier name | Snowline location | | | CReSIS data frames (YYYYMMDD_SS_FFF-FFF) * |
|---|---|---|---|---|
| | Latitude (°N) | Longitude (°W) | Elevation | |
| Kaskawulsh | 60.7501 | 139.3066 | 2062.00 | 20210510_03_001-006 |
| | 60.6970 | 139.3633 | 2105.55 | 20210510_03_039-041 |
| Steller | 60.5913 | 143.4026 | 1325.90 | 20210512_02_003-005 |
| Logan | 60.7215 | 140.1521 | 2161.84 | 20210513_02_045-046 |
| Nabesna | 61.8658 | 143.4937 | $2176.01^{+1}$ | 20210503_02_015-018 |
| E. Bagley | 60.4967 | 141.7715 | $1528.14^{+2}$ | 20210513_02_002-006 |

* In the data frame names, Y, M, D, S, F represent year, month, day, data segment, and data frames, respectively.
[+1] Compared to 2100 m in [Partington, 1998]. [+2] Compared to ~1320 m [Arcone, 1998].