# Peer review of "Snow stratigraphy observations from Operation IceBridge surveys in Alaska using S/C band airborne ultra-wideband FMCW radar"

_EGUsphere, 2022_

## Referee Comment (RC2)

**Snow stratigraphy observations from Operation IceBridge surveys in Alaska using S/C band airborne ultra-wideband FMCW radar**

Jilu Li, Fernando Rodriguez-Morales, Carl Leuschen, John Paden, Daniel Gomez-Garcia, Emily Arnold

The Center for Remote Sensing of Ice Sheets, University of Kansas, Lawrence, KS 66045, USA

5   *Correspondence to*: Jilu Li (jiluli@ku.edu.com)

**Abstract.** During the concluding phase of the NASA Operation IceBridge (OIB), we successfully completed two airborne measurement campaigns (in 2018 and 2021, respectively) using a compact S/C band radar installed on a Single Otter aircraft and collected data over Alaskan mountains, ice fields, and glaciers. We observed snow strata in ice facies, wet-snow/percolation facies and dry snow facies from radar data. This paper reports seasonal snow depths derived from our

10   observations. We found large variations in seasonal radar-inferred depths assuming a constant relative permittivity for snow equal to 1.89. The majority of the seasonal depths observed in 2018 were between 3.2 m and 4.2 m, and around 3 m in 2021. We also identified the transition areas from wet-snow facies to ice facies for multiple glaciers based on the snow strata and radar backscattering characteristics. Our analysis focuses on the measured strata of multiple years at the caldera of Mount Wrangell to estimate the local snow accumulation rate. We developed a method for using our radar readings of multi-year

15   strata to constrain the uncertain parameters of interpretation models with the assumption that most of the snow layers detected by the radar at the caldera are annual accumulation layers. At a 2004 ice core and 2005 temperature sensor tower site, the locally estimated average snow accumulation rate is ~2.89 m w. e. a-1 between the years 2002 and 2021. Our estimate of the snow accumulation rate between 2005 and 2006 is 2.82 m w. e. a-1, which matches closely to the 2.75 m w. e. a-1 inferred from independent ground-truth measurements made the same year. We also found a linear increasing trend of 0.011 m w. e.

20   a-1 per year between the years 2002 and 2021. With this trend, we extrapolated the snow accumulation back to 1992 and obtained an average accumulation rate of 2.74 w. e. a-1 between the years 1992 and 2004, which agrees well with the value of 2.66 w. e. a-1 for the same period determined from the ice core data retrieved at the caldera in 2004. The results reported here verified the efficacy of our method, its assumption, and the interpretation models.

**1 Introduction**

25   Glaciers outside Greenland and Antarctica play an important role in the Earth's ecosystem. They respond to climate dynamics in a unique way, potentially contributing to global sea level and having an impact on regional hydrology and economy. According to a recent report [WCRP Global Sea Level Budget Group, 2018], these glaciers are the second largest contributor to sea-level rise, after ocean thermal expansion, contributing 21 percent of the global mean sea-level rise during the period between 1993 and 2018. Another study claims that global glaciers are increasingly losing ice mass since the twenty first
* * *
[Figure]

**Page: 2**

Author: hpm    Subject: Highlight    Date: 9/18/22, 5:28:00 PM
Consider just "accumulation" since they aren't really different components

[revised manuscript text omitted]

* * *
**Page: 3**

Author: hpm   Subject: Cross-Out   Date: 9/18/22, 5:32:03 PM

Author: hpm   Subject: Inserted Text   Date: 9/18/22, 5:32:39 PM
with

Author: hpm   Subject: Cross-Out   Date: 9/18/22, 5:33:11 PM

Author: hpm   Subject: Cross-Out   Date: 9/18/22, 5:33:14 PM

Author: hpm   Subject: Highlight   Date: 9/18/22, 5:34:19 PM
consider stating along-track resolution, which is probably more useful to the reader than the size of the data.

Author: hpm   Subject: Highlight   Date: 9/18/22, 5:35:49 PM
2019?

Author: hpm   Subject: Highlight   Date: 9/18/22, 5:38:00 PM
Isn't this more of a factor of the pulse length, rather than the bandwidth? If the sweep rate is the same, then I can see that changing the bandwidth changes the maximum altitude?

[revised manuscript text omitted]

---

## Author Comment (AC1)

Dear Dr. Harcourt,

Thank you very much for reviewing the manuscript. We have greatly improved the paper by addressing your valuable comments. In this document, we list our replies (in black) to each of your comments/questions (in blue) and changes (in red) that are going to be integrated into the revised manuscript:

**Overview**

This paper uses radar measurements from OIB campaigns in summer 2021 over Alaska to 1) track annual snow layers in radargrams, 2) estimate the snow accumulation rate over Mt Wrangell, and 3) analyse snow strata within ice facies. The authors develop a radar age-depth model from Clark et al. (1989) to quantify the two-way travel time of the radar wave through the snow and ice, and constrain the associated parameters using a cost function that aims to minimise the difference between two snow depositional ages to annual increments (i.e. 1 year). The modelled age-depth relationship fits the derived data sets very well and hence provides confidence that the subsequent estimated accumulated rate is sufficiently accurate, with the caveat that local surface processes such as wind redistribution are not completely accounted for. The key result from a glaciological perspective is shown in Figure 8 which shows increasing accumulation rates between 2004 to 2021.

Thank you for your overview. We agree with you that the key results are shown in Figure 8.

**General Comments**

The paper is well-written overall and provides a very detailed account of the methods used to constrain the parameters in the age-depth model and the subsequent extraction of key variables such as annual snow accumulation rate. The results will be of significant interest to glaciologists and hydrologists interested in understanding glacier mass balance and its impacts on catchment hydrology. Whilst the technical details of the paper are well described, the glaciological interpretation of the data set is under-developed. In particular, I think the paper would benefit from a discussion about surface mass balance processes and how these have changed over time e.g., what are the processes underpinning the increasing accumulation rates in Figure 8. Are there any regional SMB measurements that you can compare to? I've noted some relatively minor technical corrections below which are mostly areas of clarification. If the authors can integrate these suggestions into a revised manuscript, I believe the paper will be ready for publication.

Thank you for your positive general comments and insightful suggestions. Based on your suggestions, in the revised manuscript we provided discussions about the surface mass balance processes and comparisons with the regional SMB and accumulations derived from the Modèle Atmosphérique Régional (MAR) regional climate model data. We worked on this with Dr. Xavier Fettweis who is an expert in the regional atmospheric climate model. Both Dr. Xavier Fettweis and Ibikunle Oluwanisola are now coauthors of this paper for their important contribution to MAR data analysis. We added the following at the end of Section 3.2:

[revised manuscript text omitted]

We accordingly revised L19 in the abstract as:

Additionally, we discovered a linear increasing trend between the years 2003 and 2021 of 0.011 m w. e. a[-1], which was supported by comparisons with the surface mass balance (SMB) derived for the same period from the regional atmospheric climate model MAR (Modèle Atmosphérique Régional). According to MAR data, which show an increase of 0.86°C in this area for the period of 2003-2021, the linear upward trend is associated with the increase of snowfall and rainfall events because of global warming. The findings of this study confirmed the viability of our methodology, as well as its underlying assumptions and interpretation models.

We also accordingly revised L357-359 in Section 4 for summary and conclusions:

The noteworthy discovery of the linear rise trend in accumulation rate between the years 2003 and 2021 as a result of global warming was corroborated by comparisons with the SMB derived for the same period from the MAR model. The findings of this investigation confirmed the validity of our technique and the assumptions and interpretation models it was based on. Future research may extend these findings throughout the entire caldera for the geographical pattern of snow accumulation utilizing gridded observations of strata.

**Technical Corrections (References to line numbers in preprint)**

*Abstract*

L9-L10: This sentence should come straight after your introductory line. Then you can launch into a description of your findings including observing snow strata in ice facies.

As suggested, we reordered L8-L13 as:

This paper reports seasonal snow depths derived from radar data. We found large variations in seasonal radar-inferred depths with multi-modal distributions assuming a constant relative permittivity for snow equal to 1.89. About 34% of the depths observed in 2018 were between 3.2 m and 4.2 m, about 30% of the depths observed in 2021 were between 2.5 m and 3.5 m. We observed snow strata in ice facies, wet-snow/percolation facies and dry snow facies from radar data and identified the transition areas from wet-snow facies to ice facies for multiple glaciers based on the snow strata and radar backscattering characteristics.

*Introduction*

L25: Not sure what is meant by 'Earth's ecosystem'. Maybe just 'the Earth's climate system'.

Revised as suggested:

Glaciers outside Greenland and Antarctica play an important role in the Earth's climate system and respond rapidly to changes in climate which impacts regional hydrology and the local economy.

L25-26: Suggest change to: 'in the Earth's climate system and respond rapidly to changes in climate which impacts regional hydrology and the local economy.'

Revised as suggested, see our reply above.

L29-L33: This is a long sentence and can be shortened. Focus on the global trend and then specify exactly the mass loss from Alaskan glaciers as an example. Where possible, avoid lots of clauses as it breaks up the flow of the sentence.

This long sentence was revised into two short sentences as suggested.

Another study claims that global glaciers are increasingly losing ice mass since the twenty first century and contributed 6 to 19 percent of the observed acceleration of sea-level rise during 2000-2019; the mass loss of Alaska glaciers was the biggest contributor and accounted for 25 percent of the global glacier mass loss compared to the second largest contributor, glaciers of the Greenland periphery, with 13 percent [Hugonnet et al., 2021].

L34: Start new sentence here: '…Hill et al., 2015).The changes in glacier discharge…'

Revised as suggested.

L35: 'home to important'

Revised as suggested.

L39: Maybe spaceborne?

Revised as suggested.

L41: 'However,"

Revised as suggested.

L51-53: Worth stating that ground-based measurements are also used for satellite validation of snow products.

Revised as suggested:

Ground-based measurements are used to validate both airborne and satellite observations and data products, and airborne data can also be used to validate satellite observations and data products [Lindsay, et al., 2015; Ramage, et al., 2017; Largeron, et al., 2020; Jeoung et al., 2022].

L55: 'at a glacier-scale'

Revised as suggested.

L58: 'within temperate firn'

Revised as suggested.

L63: 'with a 6-GHz'

Revised as suggested.

L64: Reference Figure 1 here

Revised as suggested.

Revised as suggested.

L71: 'across a broader spatial region than compared to the 2018 campaign (Li et al. 2019)'

Revised as suggested.

*Data collection and processing*

L79: 'over 8 days covering 5115 linear km'

Revised as suggested.

L80-88: It would be better to briefly describe and LiDAR system and discuss the radar antenna installation in a little more detail rather than referring to a previous paper. A table of critical radar system parameters would also be useful.

We added the following information as suggested:

Table 1 lists the key system parameters of the CReSIS's compact FMCW snow radar system. The details of the on-board LiDAR from the University of Alaska, Fairbanks can be found in [Johnson et al., 2013]. The snow radar's transmit antenna was installed in a protective dielectric radome under the noise of the aircraft, and its receive antenna and the LiDAR were installed in a circular port located in the aft area of the aircraft.

Table 1: System parameters

| Parameter | Value | Units |
|---|---|---|
| Weight | 35 | lb |
| Dimensions (WHD) | 14.5,9,7 | inch |
| Frequency band | 2-6 | GHz |
| Pulse duration | 250 | Ms |
| Pulse repetition frequency | 4 | kHz |
| Peak transmit power | 1 | W |
| Range resolution | 3.75 (free space) | cm |
| Hardware averages | 16 | |
| Antenna type (TX/RX) | Horn | |
| bandwidth | 2-18 | GHz |

| | | |
|---|---|---|
| beamwidth | 86-19/52-9 | degrees |
| gain | 7-13/9-23 | dBi |
| A/D converter | 14 | bit |
| Sampling rate | 125 | MSPS |

L81: 'altitude above ground level (AGL)'

Revised as suggested.

L83-84: To understand this the reader would also need to know the ADC sampling frequency, which can go into a table of parameters.

Revised as suggested (see Table 1).

L85: State the vertical resolution before and after changing the bandwidth.

Revised as suggested.

L90: Change brown to colour to distinguish from red; black might work?

Changed as suggested.

L92-93: State spatial coverage in km2?

We rewrote L90-94 to address your comment on Fig. 1, and the spatial coverage was stated accordingly in km^2. See our reply to the comment on Fig. 1.

L99: What was the magnitude of the correction applied to the radar system delay?

The corrections applied to the radar system delay were 0.064 $\mu s$ and 039 $\mu s$ in 2018 and 2021, respectively. We added this information in the revised manuscript.

L101-104: It would be helpful to provide a little more detail on these processing steps e.g., general outline of how the processing performed, performance improvement and the reason for using each step. Does the order matter? Similar to the deconvolution, were any of these steps applied differently to previous campaigns? Results analysis and discussions

We added a processing flowchart in Figure 2 and greatly expanded the discussion with details:

Figure 2 shows the data processing flowchart with 8 main steps:

1) The GPS and radar data were synchronized using the UTC time stored in the raw radar data files. The accurate longitudes, latitudes and elevations of the radar phase center along the flight path were computed with the position information of the radar and GPS antenna and the information of aircraft attitudes provided by the onboard IMU (Inertial Measurement Unit) system. Each trace of the raw radar data was tagged with the longitude and latitude of the radar antenna's phase center as its geolocation, and the elevation of the antenna's phase center was used as the zero reference for the two-way travel time (TWTT) from the aircraft to the surface.

2) The coherent noises were automatically tracked by finding the near-DC component in slow-time and were removed by subtraction. The coherent noises were caused by the feedthrough signal due to antenna coupling and undesired spurious signals generated from active microwave components of the radar system. These noises would reduce the signal-to-noise ratio (SNR), interfere with surface tracking and deconvolution if were not removed.

3) A fast-time FFT (Fast Fourier Transform) was applied trace by trace with a Hanning window to reduce sidelobes. This step, analogous to pulse compression, obtained the target response as a function of range.

4) A deconvolution filter was applied after the fast-time FFT to further reduce sidelobes and the range resolution degradation due to any other system artifacts, such as signal reflections between radar hardware components, filter's nonlinear group delay, the digital chirp's amplitude variations and frequency nonlinearity. Minimizing sidelobe level is important because sidelobes from strong interfaces could be misinterpreted as snow layers or mask weak reflections from real interfaces. The implemented deconvolution filter was an inverse filter of the radar system impulse response which was derived using specular returns from electrically smooth surface such as the calm-water surface of a lake.

5) The coherent integration was performed by stacking data traces together with the averages. This process was an unfocused SAR (Synthetic Aperture Radar) processing to improve the SNR. It included hardware and software stacking. The hardware stacking was implemented in the radar's digital system and reduced the volume size of the recorded data. The software stacking was carried out after the deconvolution in data processing. The incoherent integration was carried out after the coherent software stacking by taking the average of

the squared data of several traces. Incoherent integration reduced the signal fading effects and the data size of the final radar echogram. The number of traces in the coherent hardware integration was 8 and 16 in 2018 and 2021, respectively. The number of traces in the coherent software and incoherent integrations was 2 and 5 respectively in both 2018 and 2021. The PRF (Pulse Repetition Frequency) was 4000 Hz and 6250 Hz in 2018 and 2021, respectively. The combined coherent and incoherent integrations determined the spatial sampling frequency along the flight path and the along-track resolution depended on the aircraft velocity and the effective PRF which is 50 Hz and 39.0625 Hz in 2018 and 2021, respectively. At the typical velocity of 50 m/s during the surveys, the along-track resolution was 1m and 1.28 m in 2018 and 2021, respectively.

6) The surface was automatically tracked at this step using a threshold method. The automatic tracking usually picked the surface nicely except at the locations where the Nyquist zone changed, or the surface elevation changed very rapidly between narrow valleys. In the latter case the backscattering from both sides appeared in the leading edge of the surface and affected the threshold tracker. At these locations we corrected the surface tracking semiautomatically in our picker using manual control points.

7) The data was elevation compensated with accurately tracked surface to remove large aircraft elevation changes for effective data truncation, display radar echograms and post radar images. Two mostly used compensation options were WGS-84 elevation compensation and depth elevation compensation. The radar echogram or image was showing the real surface topography in WGS-84 datum after the WGS-84 elevation compensation. The surface was flattened after the depth elevation compensation to better display the depth between snow layers. The depth elevation compensation was implemented by using a low pass filter to get a smoothed version of the tracked surface in radar echograms, the smoothed surface was then used as the zero-depth reference and the radar echograms were normalized to this reference. The high-frequency texture of the surface was therefore kept after the surface flattening.

8) The final processed radar data and images were generated according to selected elevation compensation method.

The same processing steps and parameters were used in processing the 2018 and 2021 datasets except the above-mentioned different bandwidth, hardware stacking and PRF settings. More

discussions about the data processing procedures can be found in [Panzer et al. 2013; Yan et al., 2017].

[Figure]

**Figure 2: Flowchart of data processing main steps**

L112: "above sea level"

Revised as suggested.

L113: 'focus on the analysis of

Revised as suggested.

L114: 'discuss observations along the transition from the accumulation to the ablation zone along

Revised as suggested.

L122: I assume by 'flattening' you mean normalised to surface elevation? If so, was this from the lidar data?

No, the "flattening" did not use lidar data. See the above explanations for processing step 8.

L124-126: Both years have multi-modal peaks largely ranging between 1-6 m. Better to state this and the means of each individual distribution. This would also reveal the lack of a third peak in the 2021 data. Why might this be? More melt?

We revised L124-126 according to the suggestion and explained the lack of a third peak in the 2021 data:

Both years have multi-modal peaks largely ranging between 1-6 m. For the 2018 data, the mean values of the three distributions are around 1.2 m, 3.7m and 5.5 m. For the 2021 data, the mean values of the two distributions are around 1.1 m and 3 m. The third distribution in 2018 were mainly from thick seasonal snow along Logan Glacier and the upper Hubbard Glacier where we did not fly over these locations in 2021 (See Fig 1(c)).

We revised L10 in the abstract:

We found large variations in seasonal radar-inferred depths with multi-modal distributions assuming…

We also revised L348-340 in section 4 for summary and conclusions:

The seasonal snow depths have multi-modal distributions. About 34% of the depths observed in 2018 were between 3.2 m and 4.2 m, about 30% of the depths observed in 2021 were between 2.5 m and 3.5 m.

L133: What month were the 1994 measurements taken and are you able to quantify differences in air temperature between that study and this one?

According to [Arcone, 2002], the 1994 measurements were taken in early summer. The specific month were not stated, but I guess it should be in June, the first month of the summer in Alaska. According to the regional atmospheric climate model MAR, the average temperature in June was -0.85 degrees Celsius in 1994; the average temperature in May was -3.9 and -3.4 degrees Celsius in 2018 and 2021, respectively.

L139: Change 'massive' to 'large'

Revised as suggested.

L141: 'researchers have been drawn to study glacier-volcano interactions

Revised as suggested.

L144: 'are also both'

Revised as suggested.

L145: 'covers a 4.2 km by 2.7 km area.'

Revised as suggested.

L155: 'subsurface layers'

Revised as suggested.

L169: 'shows a plot of the flight line'

Revised as suggested.

L180: It might be beneficial to have a short sentence explain what is meant by an 'interpretation model'.

We added the following to explain the "interpretation models" after Eq. (7):

We refer to the empirical density-depth profile, the snow density-permittivity profile, and the physical processes and assumptions underlying the equations as interpretation models.

182-189: It's very hard to differentiate the notation for density and pressure. Maybe change the notation for pressure to capital P for readability?

We changed the notation for pressure to capital P as suggested.

L210: I agree with the assumption of steady-state conditions. Maybe also state that based on S3 there is also a skew towards more positive differences which could imply more snow accumulation in winter 2021.

Actually, the skew towards more positive differences implies less snow accumulation in 2021. We added the following at the end of L210:

Based on Fig. S3(c), there is a screw towards more positive differences which implies less snow accumulation in 2021. This is supported by MAR outputs which shows the surface mass balance was 3.1 m w. e. and 2.7 m w. e., respectively in 2018 and 2021.

L227: As far as I can see you haven't stated how the layers were picked – manual, semi-automatic or automatic?

We added the following at the end of L228:

The snow layers were tracked using semiautomatic methods through the GUI (Graphic User Interface) of our picking tool. Control points were manually placed along each layer and one of the automatic linear interpolation, snake and Viterbi trackers was selected to best track the layer between these control points efficiently. In most cases the Viterbi tracker best and efficiently tracked the layers [Berger et al.,2019].

L228: What density values were used to calculate the permittivity, kg/m3?

The density values were not directly used to calculate the effective permittivity. We added Eq. (11) in the revised manuscript to explain how it was calculated. According to Eq. (7), the density values are 823.53 kg/m^3 and 847.61 kg/m^3 for effective permittivity values of 2.89 and 2.96 respectively. L288-289 were revised as:

The effective relative snow permittivity $\varepsilon_{r\_eff}$ in Table 3 is calculated as:

$$\varepsilon_{r\_eff} = (\frac{c\, t_{z\_max}}{2D_{max}})^2 \tag{11}$$

where $t_{z\_max}$ is the two-way travel time from the surface to the deepest layer at the depth of $D_{max}$ observed by the radar.

L229: 1.127 km east, west, north or south?

Revised as "1.127 km southeast of the ice core site".

L251: 'values of the cost'

Revised as suggested.

L257-258: Exactly how is the value of J applied to calculate the depositional ages of the tracked layers?

As the cost function, J was minimized to determine $b_w$ and $\Delta_0$, two parameters needed to solve Eqs. (1)-(8) for the depositional ages of the tracked layers. We added the following in L258:

The closer J is to 1, the more the tracked layers are annual accumulation layers. J increases when there are intra-annual layers tracked. Because we counted dispositional ages from the surface when the data was collected, there might be a constant offset if the first annual layer was not formed one year ago. However, this offset will not affect the annual accumulation rate estimation.

L259-261: Not entirely clear why these are accumulation layers – they broadly fit into the sequence of annual integer increments…

See our comments above.

L269: "Therefore, our purpose" (i.e. because of the shift identified in the previous paragraph, only accumulation rates can be determined)

Revised as suggested.

L279-309: These are interesting results and their glaciological intepretations should be assessed further. Why is there a rising trend in accumulation? How does this relate to glacier mass balance? Is there any evidence for melt on ice internal layers and radar backscatter? It's worth highlighting in this section that you are interpretating radargrams from the dry snow facies to illustrate that melt layers are unlikely to be present.

We have revised L279-299 and answered most of the above questions (see our replies to your general comments). We added the following in L302 after "dry snow faces":

(for instance, the two research sites in Section 3.2 near Pit 5 in [Benson, 1968] are on the dry-snow line and represent dry snow facies since we did not observe internal layer melt from radar echograms).

L327-332: This description would benefit from some annotations of Figure 10a, particularly highlighting the broad locations of the facies.

We added annotations for snowline, previous summer layer, multi-year layers, ice facies and wet snow-faces in now Fig. 11(a).

[Figure]

*Figures*

Figure 1: A little difficult to see the flight lines. Could you have a small inset panel for the region and then extent indicators showing the two main regions surveyed?

We replaced Fig. 1 with Fig.1(a), (b), and (c), and rewrote L90-94 accordingly; we also referred to Fig. 3(a) and (b) for detailed flight lines at Mount Wrangell (WR) and Mount Bona (BO) summits:

The two survey regions A and B in Alaska are shown in Figure 1(a) on the hillshade map using the geographic coordinate system NAD83. A is a 4500 km$^2$ area that was only surveyed in 2018. The primary region, B, is 83,200 km$^2$, and it was surveyed in 2018 and 2021. The locations of Ultima Thule Lodge are indicated by the red start. The flight paths for areas A and B in both years are shown in Figures 1(b) and (c), respectively. The campaign's flight lines for 2018 are colored green and red, while those for 2021 are colored black and blue. There are many areas of B that were examined in both campaigns that overlap. The two-letter annotations, which use the first two letters in their names, identify the locations of the glaciers and mountains discussed in this text.

[Figure]

**Figure 1: Coverage maps of Snow Radar data from the OIB surveys in Alaska: (a) locations of survey area A and B; (b) Flight lines over A, surveyed in 2018 only; (c) Flight lines over B, surveyed both in 2018 and 2021. Green and red colors represent the locations where the Snow Radar collected data in 2018; flight lines in black and blue colors represent the locations where the Snow Radar collected data in 2021; specifically, the red and blue lines represent the locations where snow layer or snow-ice interface or snow-rock interface below the surface were observed by the compact Snow Radar. The red star marks the location of Ultima Thule Lodge. The two-letter annotations indicate the locations of some glaciers and mountains using the first two letters in their names. Refer to Fig. 3(a) and (b) for detailed flight lines at Mount Wrangell (WR) and Mount Bona (BO) summits. The hillshade map was provided by Dr. C. Larsen.**

Figure 3: A legend stating what the blue and red dots represent would be helpful.

Added legends as suggested:

[Figure]

Figure 4: Could you also annotate the location of the surface for clarity?

We added annotations for the surface:

[Figure]

Figure 9: Better to state the elevation of the snow surface in panel d.

As suggested, we changed the y-axis label as "Surface elevation (m)".

---

## Author Comment (AC2)

Dear Dr. Marshall,

Thank you very much for reviewing the manuscript. We have greatly improved the paper by addressing your valuable comments. In this document, we list our replies (in black) to each of your comments/questions (in blue) and changes (in red) that are going to be integrated into the revised manuscript:

This is a valuable, well executed study, demonstrating the ability of ultra-wideband airborne FMCW radar to measure annual accumulation layers on mountain glaciers, as well as to distinguish between dry snow and percolation/wet snow facies, and ice facies. The methodology is well described, and the combination of the radar observations with a one-dimensional physically based depth-age model is solid. The authors show good agreement between the radar-derived accumulation rate estimates and some limited field observations, lending confidence to the approach. The results are significant both in terms of the radar observation advances, and the implications of the measured accumulation rates. This will be an important paper, and just needs a bit more detail and sensitivity analysis before publication.

Thank you for your positive overview comments. To improve the paper, we added more details about data processing, comparisons with the results from the regional atmospheric climate model MAR (Modèle Atmosphérique Régional) and sensitivity analysis, see our replies to the general comments 1), 4) and 5) below.

General comments:

1) A few more details on the processing approach (i.e., what FFT windowing method was used, what kind of horizontal filtering was applied) would be helpful to add.

We added a processing flowchart in Figure 2 and greatly expanded the discussion with details:

Figure 2 shows the data processing flowchart with 8 main steps:

1) The GPS and radar data were synchronized using the UTC time stored in the raw radar data files. The accurate longitudes, latitudes, and elevations of the radar phase center along the flight path were computed with the position information of the radar and GPS antenna and the information of aircraft attitudes provided by the onboard IMU (Inertial Measurement Unit) system. Each trace of the raw radar data was tagged with the longitude and latitude of the radar antenna's phase center as its geolocation, and the elevation of the antenna's phase center was used as the zero reference for the two-way travel time (TWTT) from the aircraft to the surface.

2) The coherent noises were automatically tracked by finding the near-DC component in slow-time and were removed by subtraction. The coherent noises were caused by the feedthrough signal due to antenna coupling and undesired spurious signals generated from

active microwave components of the radar system. These noises would reduce the signal-to-noise ratio (SNR), interfere with surface tracking and deconvolution if were not removed.

3) A fast-time FFT (Fast Fourier Transform) was applied trace by trace with a Hanning window to reduce sidelobes. This step, analogous to pulse compression, obtained the target response as a function of range.

4) A deconvolution filter was applied after the fast-time FFT to further reduce sidelobes and the range resolution degradation due to any other system artifacts, such as signal reflections between radar hardware components, filter's nonlinear group delay, the digital chirp's amplitude variations and frequency nonlinearity. Minimizing sidelobe level is important because sidelobes from strong interfaces could be misinterpreted as snow layers or mask weak reflections from real interfaces. The implemented deconvolution filter was an inverse filter of the radar system impulse response which was derived using specular returns from electrically smooth surface such as the calm-water surface of a lake.

5) The coherent integration was performed by stacking data traces together with the averages. This process was an unfocused SAR (Synthetic Aperture Radar) processing to improve the SNR. It included hardware and software stacking. The hardware stacking was implemented in the radar's digital system and reduced the volume size of the recorded data. The software stacking was carried out after the deconvolution in data processing. The incoherent integration was carried out after the coherent software stacking by taking the average of the squared data of several traces. Incoherent integration reduced the signal fading effects and the data size of the final radar echogram. The number of traces in the coherent hardware integration was 8 and 16 in 2018 and 2021, respectively. The number of traces in the coherent software and incoherent integrations was 2 and 5 respectively in both 2018 and 2021. The PRF (Pulse Repetition Frequency) was 4000 Hz and 6250 Hz in 2018 and 2021, respectively. The combined coherent and incoherent integrations determined the spatial sampling frequency along the flight path and the along-track resolution depended on the aircraft velocity and the effective PRF which is 50 Hz and 39.0625 Hz in 2018 and 2021, respectively. At the typical velocity of 50 m/s during the surveys, the along-track resolution was 1m and 1.28 m in 2018 and 2021, respectively.

6) The surface was automatically tracked at this step using a threshold method. The automatic tracking usually picked the surface nicely except at the locations where the Nyquist zone changed, or the surface elevation changed very rapidly between narrow valleys. In the latter case the backscattering from both sides appeared in the leading edge of the surface and affected the threshold tracker. At these locations we corrected the surface tracking semiautomatically in our picker using manual control points.

7) The data was elevation compensated with accurately tracked surface to remove large aircraft elevation changes for effective data truncation, display radar echograms and post radar images. Two mostly used compensation options were WGS-84 elevation compensation and depth elevation compensation. The radar echogram or image was showing the real surface topography in WGS-84 datum after the WGS-84 elevation compensation. The surface was flattened after the depth elevation compensation to better display the depth between snow layers. The depth elevation compensation was implemented by using a low pass filter to get a smoothed version of the tracked surface in radar echograms, the smoothed surface was then used as the zero-depth reference and the radar echograms were normalized to this reference. The high-frequency texture of the surface was therefore kept after the surface flattening.

8) The final processed radar data and images were generated according to selected elevation compensation method.

The same processing steps and parameters were used in processing the 2018 and 2021 datasets except the above-mentioned different bandwidth, hardware stacking and PRF settings. More discussions about the data processing procedures can be found in [Panzer et al. 2013; Yan et al., 2017].

[Figure]

**Figure 2: Flowchart of data processing main steps**

2) The sensitivity of the results to the chosen permittivity should be evaluated. A relative seasonal snow permittivity of 1.89 was used, based on work on other glaciers - how does this compare to this study site? If field measurements are not available from the time of these flights, how much do results change for a conservative range of seasonal snow density/permittivity?

We added the following at the end of Section 3.1:

There are not many large-scale radar snow measurements over Alaska glaciers, yet they are very important for studies on regional hydrology and mass balance. The goal here is to present the spatial distributions of the seasonal snow our radar has detected. We kept track of the seasonal snow cover in our datasets to facilitate these studies. However, the focus of this work does not extend to these studies, which necessitate a detailed understanding of the snow density profile and its tempo-spatial fluctuations.

3) How were the layers picked? Manually? Semi-automatically with control points? Automatically? This needs more detail.

We added the following at the end of L228:

The snow layers were tracked using semiautomatic methods through the GUI (Graphic User Interface) of our picking tool. Control points were manually placed along each layer and one of the automatic linear interpolation, snake and Viterbi trackers was selected to best track the layer between these control points efficiently. The Viterbi tracker typically tracked the layer most effectively [Berger et al., 2019].

4) The model appears to produce a depth-age scale, but also a depth-density result. How does this compare to the ice core data? The authors state that although the model was tuned to Greenland, it represents the firn well - it would be useful to show this with field observations from this site, which should be available from the ice core. For example, Greenland gets a huge amount of wind influence, and the authors state that the main site is not wind effected. It's possible the assumed surface density in the model, for example is a bit too high.

The Benson's density-depth profile describe by Eq. (2) was based on the ground measurements in Greenland. Based on the measurements he later performed for the temperature, density, hardness, and stratigraphic profiles in the Mount Wrangell caldera [Benson, 1968], he stated that "facies parameters calculated for the summit area of Mount Wrangell (4,000 to 4,300 m. at 62 °N.) compare well with the same parameters near the dry-snow line on the Greenland Ice Sheet".  In [Benson, 1968], the averaged snow density for the top 5 m was 390,398,390,360 kg/m^3 at four locations Pit 2, Pit 3, Pit 4 and Pit 5, respectively. The average of these locations is 384.5 kg/m^3. Location Pit 5 is very close to study sites, therefore with more weight on Pit 5, the value used as the surface density in this study, 377.36 kg/m^3, might be reasonable for the average density of the top 5 meter. This may need to be scaled down more for the density at the surface. However, as the sensitivity study below shows, the accumulation estimation would not have significant changes.

The surface density value is 317.5kg/m^3 according to MAR data, we used this value to evaluate the accumulation estimation's sensitivity to the surface density values. We added the following after the paragraphs added for comparisons with MAR results (see our replies to the general comment 5):

According to MAR data, the surface density in Mount Wrangell's caldera is $317.50\ kg/m^3$. This figure is 16% less than the value we used in the study, $377.36\ kg/m^3$. The models' and accumulation estimations' sensitivity to the surface density values was therefore further evaluated. The discrepancies in the density-depth profiles for the two surface density values are depicted in Figure 9(a). As seen in Fig. 9(b), as depth is increased, the projected depositional ages for the tracked layers would get less due to the lower surface density. As opposed to 18.6 years for $377.36\ kg/m^3$, the age of the deepest monitored layer is 17.10 years for $317.50\ kg/m^3$. The variations between the annual accumulation estimates are compared in Figure 9(c). Although there are some variations in the annual accumulation rate within a given year, the linear increasing trend is nearly the same ($0.011\ m\,w.e.\,a^{-2}$ for $317.56\ kg/m^3$ against $0.012\ m\,w.e.\,a^{-2}$ for $377.36\ kg/m^3$). This makes sense given that, for a lower snow density, the snow mass likewise decreases as the age difference between two snow layers narrows. As a result, we deduced that the linear upward trend in the annual accumulation rate seen between 2003 and 2021 is not affected much by the surface density.

[Figure]

(a)

(b)

(c)

**Figure 9: Depth-density profiles (a), Snow layer depositional ages (b), and estimated annual accumulation rates (c) for two different surface density values.**

5) This is the most important general comment -- the interpretation of the linear increase in accumulation rate is the most important result from a glaciological/snow science point of view but needs a bit more work to test this trend, and explain why it might be happening. Is this linear increase over the past several decades expected based on regional climate models? Other glacier observations? Even if we had a linear increase the last 2 decades, why would we expect this to be the case in the decade previous (which was used to extrapolate to the early 1990s)?

I'm honestly a bit worried that this trend is caused by an assumption of constant density, or some artifact in the model. Can you back this linear increase in accumulation up with any other regional data from field observations or models? Or how about the ice core - does that show a linear increase in accumulation rate? I would expect the chemistry in the ice core would have resulted in a depth-age scale, so accumulation time series should be available from that?

Related to this - your model assumes a steady state - that the accumulation rate is balanced by the melt/densification and flow divergence. You show this with the little change in surface elevation between 2018 and 2021. Then what does the increasing accumulation over time then imply? Greater basal melt to balance this increase? Or a volume flux or densification rate increase with time?

Is it possible that a bias in the densification model is leading to the linear increase in estimated accumulation? I think a sensitivity analysis is needed here, along with error bars for the plot of accumulation vs time.

We addressed this most important general comment by working with Dr. Xavier Fettweis who is an expert in the regional atmospheric climate model MAR (Modèle Atmosphérique Régional). Both Dr. Xavier Fettweis and Ibikunle Oluwanisola are now coauthors of this paper for their important contribution to MAR data analysis. We compared our results with the surface mass balance estimates using MAR data. We found the linear increasing trend from MAR data is almost the same as what inferred from radar data between 2003 and 2021, although the MAR results have larger variations from year to year. We also found MAR results do not have this linear increasing trend between 1990 and 2003.

We thought the only ground truth from the 2005 temperature sensor measurements was limited, we therefore extrapolated to 1992 based on the linear trend between 2003 and 2021 to compare with another ground truth determined from the 2004 ice core using the dating of a tephra layer from the 1992 Mount Spur eruption mentioned in [Kanamori et al., 2008]. Although the estimated average accumulation rate for the years between 1992 and 2004 is very close to the value from the ice core, we removed the extrapolation part from the revised manuscript. Fig. 8 has been accordingly revised with the comparisons with MAR results.

After analyzing the MAR data for SMB, snowfall, rainfall, melt, runoff, surface temperature, and snow densities in the first 10 m depths between 2003 and 2021 (see the table below), we came to the conclusion that the increasing accumulation over this period was associated with the increase in snowfall and rainfall events due to global warming and was primarily balanced by an increase in densification rate, with flow divergence playing a smaller role. As shown in the table and Fig. 8 (d), the snow surface temperature at the study site increased 0.86 °C. The snowfall and rainfall contributed about 88% and 12% to the SMB increase. Respectively. There might be some melt in summer. There were no runoffs, which means the melt did not contribute to SMB and the rainfall was fully retained by snowpacks. Sublimation/evaporation was small and constant over 2003-2021.

Table: SMB process at the 2004 ice core/2005 temperature sensor tower site between 2003-2021

|  | SMB (m w.e.) | Snowfall (m w.e.) | Rainfall (m w.e.) | Melt (m w.e.) | Density (0-10m) (kg/m$^3$) | Temperature (°C) |
|---|---|---|---|---|---|---|
| mean | 2.96 | 2.97 | 0.0078 | 0.0516 | 410.2 | -15.8 |
| std dev | 0.26 | 0.27 | 0.0236 | 0.0556 | 26.8 | 1.0 |
| Slope | 0.012 | 0.011 | 0.0016 | 0.0005 | 1.8 | 0.051 |
| change | 0.205 | 0.180 | 0.0270 | 0.0087 | 21.7 | 0.86 |

The following figure shows the densification rate at the 2004 ice core/2005 temperature sensor tower site computed using the interpretation models. According to this figure, most part of the densification process happened in the first 1.5 years within the first 10 m depths. 10 m is the depth where the critical pressure is reached. According to MAR, the SMB was 2.65 m. w. e., and the mean snow density for the 0-10 m in 2021 was 417.2 kg/m3. This means the SMB was equivalent to 6.25 m of snow, and the first 9.38 meter of snow in 2021 would be totally replaced after 1.5 year. The increase of the snow density of the 0-10 m is nearly statistically significant, which is supportive evidence that the snowpacks absorbed the increased rainfall and melt. This also explains why accumulation was increasing but not the surface elevation.

[Figure]

Figure: Depth and densification rate between 2003-2021 (tower site)

For model sensitivity analysis, see our replies to your general comment 4. We are more confident with the models, analysis method and results after we compared our results with MAR data. We added the following at the end of Section 3.2:

[revised manuscript text omitted]

Detailed suggestions/edits in the attached annotated PDF.

Thank you very much for the detailed suggestions/edits in the attached annotated PDF. We addressed each item as below:

L11: Consider specifying - 60%?

We revised L11 in the abstract as:

About 34% of the depths observed in 2018 were between 3.2 m and 4.2 m, about 30% of the depths observed in 2021 were between 2.5 m and 3.5 m.

We also added the above content in L126 in Section 3.1.

L19: What is the impact of constant density assumption here? I.e., if density increased during that time, how big of an effect would that have on the result?

We did not assume a constant density here. We assumed a surface density in 2021 and Eq. (2) is the density-depth profile we used. See our replies to your general comment 4 about the surface density effect on the result.

L20: Format as a^-1

As suggested, we formatted all cases in the manuscript as $a^{-1}$.

L20-21: Is there a physical reason you would expect the accumulation rate to increase linearly over this entire time period? Does the ice core show the same linear trend?

See our replies to your general comment 5.

L38: Consider just "accumulation" since they aren't really different components.

As suggested, we revised "Snow depth and accumulation" as "Snow accumulation".

L68: deleted "the" as suggested.

L71: Instead of replacing "in" with "with" as suggested, we replaced "in much broader coverage" with "across a broader region" as suggested by Dr. Harcourt.

L74: deleted "the" and "of the study" as suggested.

L79: Consider stating along-track resolution, which is probably more useful to the reader than the size of the data.

As suggested, we revised this line as "5315 linear km with an along-track resolution of ~1.3 m". Please also see processing step 5.

L81: 2019?

Revised 2018 as 2019.

L83: Isn't this more of a factor of the pulse length, rather than the bandwidth? If the sweep rate is the same, then I can see that changing the bandwidth changes the maximum altitude?

The maximum beat frequency in the first Nyquist zone is $Fs/2$, so $t\_max = 2H\_max/c = Fs/2/(BW/T)$, therefore $H\_max = (c/2) (Fs/2)/(BW/T)$, where $Fs$ is the sampling frequency, $T$ is the pulse length, $BW$ is the bandwidth and $c$ is the speed of light in free space. We did not change $T$ and $Fs$, therefore a reduction in bandwidth increased the maximum altitude. To clarify this, we added "kept the same chirp length and sampling frequency (see Table 1)" in L83. We also added Table 1 to list key radar system parameters (see Table 1).

L102: Windowed? Maybe mention what time of window used (e.g., hanning, hamming)?

Hanning window was used, see processing step 3.

L108: show

Revised "showing" as "show" as suggested.

L125-126: Possibly caused by a mid-winter rain event?

The distribution modal peaks were from seasonal snow over multiple glaciers. See the revisions on L124-126 below.

L126: a

We revised L124-126 according to the suggestions from Dr. Harcourt and explained the lack of a third peak in the 2021 data:

Both years have multi-modal peaks largely ranging between 1-6 m. For the 2018 data, the mean values of the three distributions are around 1.2 m, 3.7m and 5.5 m. For the 2021 data, the mean values of the two distributions are around 1.1 m and 3 m. The third distribution in 2018 were mainly from thick seasonal snow along Logan Glacier and the upper Hubbard Glacier where we did not fly over these locations in 2021 (See Fig 1(c)).

L131: Consider putting these in terms of permittivity rather than refractive index, to facilitate comparison.

Revised as suggested:

We note that the Bagley Ice Field is a temperate glacier, and previous investigations based on 135-MHz pulsed radar measurements in early summer 1994 determined the relative permittivity from 16.81 to 20.25 for the near-surface of Bagley Ice Field.

L135: Remove "-" around "and"

Revised as suggested.

L140: Do you mean is in the dry snow zone?

By "extends into dry snow zones", we mean the lower elevation part of the summit region may be in the upper percolation zone, and the higher elevation part is in the dry snow zone.

L141: We revised "it drew the attentions of researchers to study" as "researchers have been drawn to study" as suggested by Dr. Harcourt.

L145: is an elliptical shape

We revised "the saddle area between them are 4.2 km by 2.7 km elliptical" as "the saddle between them covers a 4.2 km by 2.7 km area" suggested by Dr. Harcourt.

L155: inert "s"

Revised "layer" as "layers" as suggested.

L156: we flew

Revised "it was" as "we flew".

L169: insert "a"

We revised "is plot" as "shows a plot" as suggested.

L176: Seems a little high, close to ice? Maybe the core was very dense. Would be worth stating the average core density.

No available ice core data to us at the same site. This was inferred from radar layer two-way travel times. Our results were confirmed with limited ground truth and the added comparisons with MAR data. We revised L288-289 to explain more clearly how the effective relative snow permittivity was computed (see our replies to your comment at this two lines).

L194: How well does this match surface density at the site? Seems appropriate for Greenland but maybe a bit high for seasonal snowpack?

Yes, it looks a bit high compared to MAR data. However, the sensitivity study did not show significant changes in accumulation estimation. See our replies to your general comment 4 for sensitivity study results and explanations.

No, it is not constant over time. This is the surface mass balance dependent on the location and time of the data collection.

We revised "implied by" as "assumed for" as suggested.

We revised "elevations" as "elevation changes" as suggested.

L221: and here the age t_a is for each of the radar horizons.

We inserted "and here the age $t_a$ is for each of the radar horizons".

We revised "to" as "southeast of" as suggested by Dr. Harcourt.

L231: How were these picked? Manually? semiautomatically? automatically?

Semiautomatically. See our reply to your general comments 3).

We revised "data lines" as "traces" as suggested.

We revised "along-track filtering" as "along-track moving average filtering".

Yes.

Revised "$\varepsilon \sim \rho$ "as "$\varepsilon = f(\rho)$ "as suggested.

Eq. (10): dz here is the depth difference between z_k and z_k-1?

We added the following in L274:

and $dz = 0.1\ m$ is the step used in integrating the differential Equations (1)-(3) and (5)-(7).

L288-289: If it was just this ratio, the units wouldn't work out. I think you mean the ratio is used to calculate the mean velocity, which is used to estimate effective permittivity?

Yes, I meant the ratio was used to calculate the mean velocity, which was used to estimate effective permittivity. We revised "The effective snow permittivity $\varepsilon_{r\_eff}$ in Table 2 is calculated as the ratio of $Dmax$ to the travel time from the surface to the deepest layer observed by the radar" as

"The effective relative snow permittivity $\varepsilon_{r\_eff}$ in Table 2 is calculated as:

$$\varepsilon_{r\_eff} = \left(\frac{c\ t_{z\_max}}{2D_{max}}\right)^2 \tag{11}$$

where $t_{z\_max}$ is the two-way travel time from the surface to the deepest layer at the depth of $D_{max}$ observed by the radar".

L317: along with pooling of liquid water at storm layer interfaces

We inserted "along with pooling of liquid water at storm layer interfaces" as suggested.

We revised "the snowline" as "where the snowline defined as the boundary".

L330: Consider using "boundary" rather than "snowline", since snowline usually refers to the boundary between snow and glacier ice.

See the revision for the comments on L324.

L342: Consider defining CReSIS somewhere in the document

Defined CReSIS under the author list as suggested.

Figure 1: Absolute location information needed. Is this UTM? What grid? Or some local coordinate system?

Revised "Figure 1 shows…on the hillshade map" as "Figure 1 shows…on the hillshade map in NAD83 geographic coordinate system".

Figure 5: respectively

Revised "reflectively" in the figure's caption as "respectively" as suggested.